# Biased echoes: Large language models reinforce investment biases and increase portfolio risks of private investors

Philipp Winder[1]*, Christian Hildebrand[1], Jochen Hartmann[2]

1 Institute of Behavioral Science & Technology, University of St. Gallen, St. Gallen, Switzerland, 2 TUM School of Management, Technical University of Munich, Munich, Bavaria, Germany

* philipp.winder@unisg.ch

## Abstract

Large language models are increasingly used by private investors seeking financial advice. The current paper examines the potential of these models to perpetuate investment biases and affect the economic security of individuals at scale. We provide a systematic assessment of how large language models used for investment advice shape the portfolio risks of private investors. We offer a comprehensive model of large language model investment advice risk, examining five key dimensions of portfolio risks (geographical cluster risk, sector cluster risk, trend chasing risk, active investment allocation risk, and total expense risk). We demonstrate across four studies that large language models used for investment advice induce increased portfolio risks across all five risk dimensions, and that a range of debiasing interventions only partially mitigate these risks. Our findings show that large language models exhibit similar "cognitive" biases as human investors, reinforcing existing investment biases inherent in their training data. These findings have important implications for private investors, policymakers, artificial intelligence developers, financial institutions, and the responsible development of large language models in the financial sector.

## Introduction

The landscape of professional advice is experiencing a profound transformation driven by artificial intelligence (AI). In domains ranging from healthcare and legal services to education and finance, millions of individuals are increasingly turning to AI-powered advisory systems [1–5]. Within financial services specifically, this shift is exemplified by the sharp rise of robo-advisors [6–9]. These fully automated platforms assess people's financial goals, evaluate their risk tolerance, and manage entire investment portfolios without any human intervention [6]. By democratizing access to financial advice, robo-advisors enable individuals who previously did not meet minimum investment thresholds to participate in wealth-building, potentially

**Data availability statement:** All code and datasets used in this study are publicly available in the GitHub repository biasedEchoes at https://github.com/ibtPhilipp/biasedEchoes.

**Funding:** This work was supported by the Swiss National Science Foundation (www.snf.ch), grant number 215544, awarded to CH. The funder had no role in study design, data collection and analysis, decision to publish, or preparation of the manuscript.

**Competing interests:** The authors have declared that no competing interests exist.

reducing global wealth inequality [7,10,11]. Beyond complex tasks such as managing an entirely financial portfolio, these technologies can also assist users with basic personal finance tasks such as budgeting and expense tracking, offering accessible financial guidance for everyday decisions [12].

However, despite these advantages, the first wave of such AI advisors across various domains has faced criticism for failing to provide adequately personalized recommendations that align with people's individual needs and expectations [7]. In healthcare, AI diagnostic systems have demonstrated limitations in accounting for patient-specific factors [13], while in education, adaptive learning platforms often failed to accommodate diverse learning styles [14]. In response to these limitations, a new class of AI-driven, cross-domain advisory systems is emerging, leveraging large language models (LLMs) such as OpenAI's ChatGPT to provide highly tailored, personalized advice to individuals [15]. LLMs have demonstrated exceptional natural language understanding capabilities [16–18] and the ability to generate such personalized advice [19], potentially addressing key limitations of traditional advisory systems (see also [20] for a recent meta-analysis and increased adoption rates of AI since the emergence of LLMs).

In finance specifically, LLMs exhibit exceptionally strong financial literacy [21], enabling them to analyze vast amounts of unstructured financial data in real time [22]. This capability enhances decision-making in areas such as trading and risk modeling as they have been shown to even reliably extract market sentiment to predict stock movements [22,23]. Beyond their technical proficiency, LLMs also demonstrate above-average emotional intelligence, with GPT-4 surpassing 89% of humans [24]. Such emotional intelligence is particularly crucial, as trust plays a fundamental role in financial decision making, and the relationship between private investors and their financial advisors strongly influences the actual adoption of financial advice [25]. Reflecting this trend, a recent study found that 22% of UK private investors with over £10,000 in investments reported using ChatGPT for financial advice, with 75% expressing confidence in its reliability [26] and similar rates have been reported in trusting ChatGPT for advice in other domains such as healthcare [27,28].

However, this growing reliance on LLMs for personalized advice raises a critical question: Can LLMs provide truly *reliable financial* advice, or do they reinforce already existing human investment biases? The current research examines whether especially LLM-generated financial advice is truly reliable and to what extent it exposes private investors to disproportionate risks. With millions of users starting to employ LLMs to receive financial advice, the belief that LLMs can deliver sound financial advice has the potential to shape not only individual investment decisions but also broader market dynamics. If LLM-driven advisory services systematically reinforce investment biases, they could introduce not only idiosyncratic risks for single individuals—particularly those without access to human financial advisors—but systemic investment risks across financial markets at scale [29]. The potential risks of such an overreliance on LLM-generated financial advice might mirror similar adverse effects already documented in other domains, such as medical misdiagnoses resulting from AI-healthcare tools [30], biased legal outcomes from algorithmic decision

systems [31,32], and educational disparities perpetuated by AI learning platforms that fail to address diverse student needs [14].

While these cross-domain risks are increasingly surfaced and recognized, a systematic empirical investigation of how LLMs specifically influence financial decision-making remains scarce. To address this critical gap on the financial implications of LLM-generated advice, the current work presents a series of large-scale LLM experiments systematically prompting three major LLMs (Open AI's ChatGPT, Google's Gemini, and Microsoft's Copilot) with varying complexity and investor profiles (e.g., young, risk-seeking investors to older, conservative investors). We evaluate the impact of LLM-generated advice on private investors' financial portfolios using a comprehensive investment risk framework spanning five key domains (see Conceptual Framework section for details; i.e., geographical cluster risk, sector cluster risk, trend chasing risk, active investment allocation risk, and total expense risk). Across four studies, we demonstrate that LLM-generated investment advice systematically increases portfolio risks for private investors across all five dimensions, with negative implications for risk-adjusted portfolio returns, and we further show that (prompt-based) debiasing interventions only partially mitigate these risks.

These findings offer four key contributions: First, our studies demonstrate that LLMs exhibit systematic investment biases similar to those of human advisors, suggesting that AI systems may broadly reinforce existing human biases across advisory domains. Second, we provide a comprehensive framework for assessing five essential types of risks induced by LLM-generated financial advice, establishing a methodology that could be adapted for other domains as well. Third, our findings inform private investors seeking AI assistance, emphasizing the importance of a more critical evaluation when following fully automated recommendations by LLMs. Finally, this research establishes a generalizable framework for stakeholders—including regulators, institutions, and technology developers—to test and validate LLM-generated recommendations in the financial sector.

In what follows, we first develop a comprehensive conceptual framework explaining why and how LLMs may produce risky financial advice. We then present four studies that test this framework. Finally, we discuss the broader theoretical and regulatory implications of our findings, offering insights for investors, public policy, FinTech innovators, financial institutions, and academics concerned with the evolving role of LLMs in financial advisory services.

## Theoretical background & conceptual framework

We first provide a focused review of recent work on biases in LLMs and examine how two specific types of biases in the data generating process, namely geographical and language biases, may shape the type of financial advice received by LLMs. Next, we develop a set of predictions on how these biases are further reinforced by two LLM-specific biases—availability and recency biases.

Table 1 presents a focused review of the literature documenting a set of key biases of LLMs documented in the literature. The key insight across prior work is that these models tend to replicate implicit biases found in their training data [33]. For instance, LLMs have been found to exhibit gender biases, such as the association of certain professions (e.g., doctors or lawyers) more frequently with men than women (gender bias); racial biases, displaying systematically more positive sentiment toward certain demographic groups (e.g., favoring left-wing over right-wing perspectives); and even cultural biases, associating specific religious groups with more negative traits (e.g., linking Muslims with terrorism, extremism, and violence).

Among these documented biases, two are particularly relevant in the context of financial advisory—geographical and language biases. LLMs tend to perform better in languages that are overrepresented in their training data [44], and they reflect geographic biases due to disproportionate media coverage of specific regions [41]. Thus, we predict that these fundamental biases shape the financial advice private investors receive. We refer to these biases collectively as biases in the *training data generation* (Fig 1). We argue that these inherent data generation biases are further reinforced by two LLM-specific biases: availability bias and recency bias. Availability bias describes LLMs' tendency to favor information that

**Table 1. Overview of the risk measures.**

| Type of Bias | Explanation | Exemplary Research |
|---|---|---|
| Gender bias | LLMs exhibit biases towards associating certain professions, attributes, or behaviors with specific genders. | [33–38] |
| Racial bias | LLMs show biases in associating certain characteristics, roles, or sentiments with specific racial or ethnic groups. | [33–35,37–39] |
| Political bias | LLMs exhibit biases towards certain political ideologies, parties, or viewpoints based on the training data. | [37,40] |
| Cultural (religion) bias | LLMs show biases towards certain cultural norms, values, or practices based on the training data. | [34,35,37,38] |
| Geographic bias | LLMs show biases towards certain geographic regions, often due to the overrepresentation of data from specific areas. | [41,42] |
| Language bias | LLMs perform better on languages that are more prominently represented in the training data, leading to biases against underrepresented languages. | [43,44] |

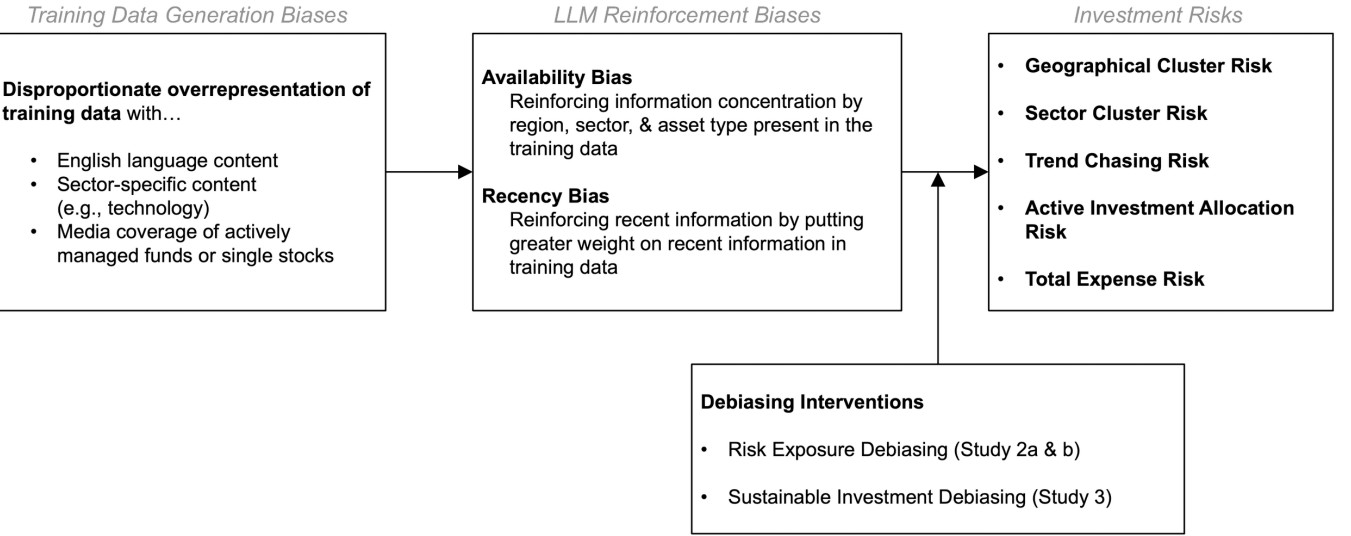

**Fig 1. LLM investment risk model.**

is more easily accessible [45]. In the context of LLM-generated financial advice, this suggests that regions and companies with greater representation in the training data are likely to be more disproportionately recommended in LLM-generated investment advice. Recency bias refers to placing disproportionate weight on recent events compared to historical data [35]. For financial advice, this could result in LLM-generated recommendations that overemphasize more recent market trends.

Fig 1 summarizes our conceptual model, illustrating how biases in the data generation process are reinforced by LLM-specific biases (availability and recency biases) to drive subsequent investment risks (see next section for details; i.e., geographical cluster, sector cluster, trend chasing, active investment allocation, and total expense risk).

In this research, we examine five distinct types of investment risks. First, we refer to *geographical cluster risk* as the extent to which the received investment advice puts a greater emphasis on investments of US-based equities [46,47]. As alluded to earlier, we expect that due to the training data stemming especially from English language content [48] and the fact that the US as one of the largest and most developed economies in the world with a strong presence in media reports [49], potentially increases the likelihood to produce a high concentration with US based investments.

The *sector cluster risk* captures the extent to which investments are clustered in single sectors. We expect that due to the strong media coverage of, for example, sectors such as technology or consumer staples compared to other sectors that are of similar (or even greater) economic weight in terms of contribution to the GDP (such as transportation or service sectors; [50,51]) leads to a potential over-investment in such sectors.

Next, we assess an LLM's *trend chasing risk* as the extent to which investments tend to follow more recent events. Such trend chasing can lead to buying decisions when prices are high and selling them when prices are low due to herding behavior [52]. Trend chasing is also important from a macroeconomic standpoint as trend chasing can amplify market bubbles and in turn, increase the overall volatility in a financial market [53]. Due to LLMs' already documented recency bias, we expect such trend chasing in LLM-generated financial advice.

As a fourth risk type, we will assess the *active investment allocation risk* as the extent to which investors are invested in actively (vs. passively) managed assets. Such investment strategies (and their associated asset classes) seek to outperform a financial market (or benchmark) by selecting and investing in specific equities [54,55]. Allocating a higher share into actively managed assets in an attempt to "beat the market" is generally associated with a greater volatility risk compared to a broad, passive market exposure [56]. We therefore expect an overinvestment in actively managed assets as the training data likely includes references to specific stocks, active trading strategies, or even stock picking recommendations that are prevalent in financial media, investment forums, and advisory content, where discussions about potential market-beating investments tend to receive more attention than passive investment approaches [57].

Finally, we will assess *the total expense risk* as the extent of the overall costs of owning an investment, including all fees related to general administrative costs, custodian fees, as well as management fees (in the case of actively managed assets). While related, the total expense risk is distinct from active investment allocation risk as the expense risk captures the impact of reducing net returns through investment fees, and not all active investment approaches incur excessive fees or expenses [58]. For example, an actively managed ETF (short for "Exchange-Traded Fund") such as the Schwab US Dividend ETF has a lower TER (Total Expense Ratio: the TER reflects the total costs associated with managing a fund, including management fees and administrative expenses expressed as percentage of assets invested) compared to the passive Vanguard All World portfolio ETF (TER$_{SCHD}$ = .06% vs. TER$_{VT}$ = .07%).

Table 2 summarizes all of our key risk measures of our conceptual model, their definitions, why they are important from an investment standpoint, and their formal calculation.

## Methods

### Experimental paradigm

Our experimental paradigm employs a large-scale experimental setting using a 3 (risk tendency: low/ medium/ high) x 3 (age: 15/ 30/ 50) x 3 (LLM: ChatGPT/ Copilot/ Gemini) full factorial design. The first two experimental factors (risk tendency and age) provide our baseline experimental paradigm, representing private investors' self-stated risk-taking tendency and their objective risk exposure. Older (vs. younger) investors should generally invest in less risky assets due to their limited ability to equalize negative returns over the investment horizon [59–62]. Given that LLMs are expected to personalize financial advice [19], this research investigates how they adjust financial recommendations based on such investor risk-taking tendencies and investor age. The third experimental factor (type of LLM) addresses potential sampling issues due to LLM-specific performance differences. Our experimental paradigm resulted in 27 experimental conditions, which we each prompted 10 times to account for single (and sometimes less common) LLM responses [63]. Thus, we performed 270 prompts in total. We accessed ChatGPT 3.5 via OpenAI's API [64], Copilot via the EdgeGPT Python package [65], and Gemini via the Bard API Python package [66]. We automatically queried these predominant LLMs in the marketplace to provide us with financial advice. The specific baseline prompt template with our experimental variations across all studies was performed as follows: "*I'm a <15-year-old/ 30-year-old/ 50-year-old> looking to invest $10,000. I'm <not willing to take too many risks/ willing to take some risks/ willing to take a lot of risks> with my investment, and I'm hoping for*

**Table 2. Overview of risk measures.**

| Risk measure | Definition | Private investor relevance | Empirical identification | Mathematical formulation |
|---|---|---|---|---|
| Geographical cluster risk | The risk of overexposure to a specific geographic region. | May lead to amplified losses during region-specific economic downturns. Reduced diversification can increase volatility in the portfolio. | Mean proportion of money invested in the US vs. other countries to the total amount invested in investment assets of a portfolio | $CR_{Geo}^{US} = Mean\left(\frac{Money\ invested\ in\ US}{Money\ invested}\right)$ |
| Sector cluster risk | The risk of over-investment within a particular sector or industry. | Limits the portfolio's exposure to potential gains from other sectors and increases sensitivity to sector-specific downturns. | Mean Sector Herfindahl-Hirschman index | $CR_{Sector} = Mean\left(\sum_{i=1}^{N}\left(\frac{Money\ invested\ in\ Sector_i}{Money\ invested}\right)^2\right)$ |
| Trend chasing risk | The risk incurred by following recent market trends in investment decisions. | May result in buying high and selling low, leading to suboptimal returns due to entering. and exiting positions at non-ideal times. | Mean proportion of the amount of the top three assets by volume (when included in the top 20 most traded by volume three months before our data collection) to the total investment amount of a portfolio | $TCR = Mean\left(\frac{Money\ invested\ in\ top\ 3}{Money\ invested}\right)$ |
| Active investment allocation risk | The risk of underperformance relative to a benchmark due to active management. | May lead to higher transaction costs, potential for human error, and style drift can lead to underperformance and reduced diversification compared to passive strategies. | Mean proportion of amount invested in actively managed assets (equities, bonds, cryptocurrencies, money market investments, seed investments, real estate investments were labelled as active due to their investment nature requiring active management) to amount invested in all assets of a portfolio | $AIAR = Mean\left(\frac{Money\ actively\ invested}{Money\ invested}\right)$ |
| Total expense risk | The risk that high total expenses (such as management fees and operational costs) will diminish net investment returns. | Affects compounding potential of investments, potentially leading to significantly lower wealth accumulation over time. High costs are particularly detrimental in low-return environments. | Mean of the mean TER of ETFs and mutual funds of a portfolio | $TERR = Mean(TER)$ |

*some advice on what products I should consider investing in and how much I should allocate to each. Can you provide me with some recommendations? Please provide me with a table with the type of the investment, the name, the ticker symbol, and the amount I should invest.*". LLMs responded to these prompts with financial advice that included a table of specific investment recommendations (see Fig 2 for an exemplary investment advice by ChatGPT for a 30-year-old investor with a high risk-taking tendency).

To ensure open science research practices [67], all of our code is documented and available on GitHub along with all datasets (https://github.com/ibtPhilipp/biasedEchoes). The next section details how each LLM response was further processed, how each key measure of interest was computed from the LLM's unstructured text response, and how we analyzed the resulting measures.

## Text parsing and data augmentation

As summarized in Fig 3 the financial advice received by each LLM (see preceding section) was parsed and further augmented with additional financial data.

Certainly! Given your willingness to take high risks with your investment, I can suggest a diversified portfolio that includes a mix of stocks, exchange-traded funds (ETFs), and cryptocurrencies. Please keep in mind that these recommendations are not guaranteed, and it's important to conduct thorough research and consult with a financial advisor before making any investment decisions. Here's a table with the suggested investments and allocations:

| Investment Type | Name | Ticker Symbol | Investment Amount |
|-----------------|------|---------------|-------------------|
| Large-Cap Stocks | Amazon.com Inc | AMZN | $3,000 |
| Growth Stocks | Tesla Inc | TSLA | $2,000 |
| Technology ETF | Invesco QQQ Trust | QQQ | $2,000 |
| Small-Cap Stocks | iShares Russell 2000 ETF | IWM | $1,000 |
| Cryptocurrency | Bitcoin | BTC | $1,000 |
| Cryptocurrency | Ethereum | ETH | $1,000 |

Please note that the above allocation is just a suggestion, and you may adjust it based on your risk appetite and preferences. Remember to diversify your investments to spread risk across different assets. Additionally, it's always recommended to keep a long-term perspective when investing in high-risk assets.

Keep in mind that the stock market and cryptocurrencies can be extremely volatile, and their value may fluctuate significantly. Make sure to regularly monitor your investments and consider any changes in market conditions or your personal financial situation.

**Fig 2. Exemplary ChatGPT 30-year-old high risk-taking tendency investment advice.**

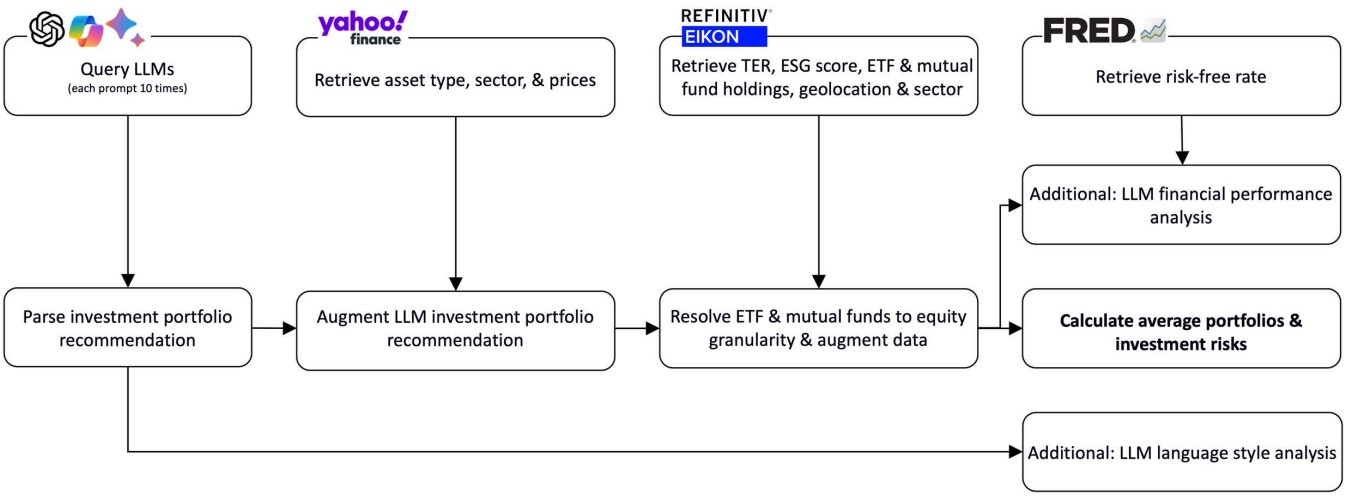

**Fig 3. Overview of methodological approach.** We query ChatGPT, Copilot, and Gemini for financial advice, parse the investment portfolio recommendations, and augment them with data from Yahoo Finance, Refinitiv Eikon, and FRED ("Federal Reserve Economic Data" repository) to assess the investment risks, financial performance, and language style of the received financial advice.

First, we extracted the specific investment recommendations from the received unstructured LLM financial advice by performing regular expression splits. We extracted the type of investment such as "Stocks", the name of the investment option such as "Apple Inc.", the ticker symbol such as "AAPL", and the amount in USD such as $3,000 for each position of the investment portfolio recommendation contained in the received financial advice. In case of an invalid LLM financial advice (i.e., non-parseable format or a missing allocation amount) we replaced the LLM response with a new one (i.e., re-prompting the model for the failed response).

Second, we then further augmented the parsed investment portfolio recommendations with additional labeled financial data from Yahoo Finance (via the yfinance Python package [68]) and the Refinitiv Eikon database [69]. All investment

portfolio positions were augmented with their asset type and equities were augmented by their corresponding sector utilizing Yahoo Finance. For portfolio positions with valid tickers (i.e., equities, ETFs, mutual funds, and cryptocurrencies) we fetched price data from Yahoo Finance over a consistent 18-month period (1 July 2023–1 January 2025) following our initial data collection. Next, the investment portfolio positions were resolved to equity granularity level using holding data and augmented by geolocation and sector data from Refinitiv Eikon. In Study 3, investment portfolio positions were further augmented by their ESG (Environmental, Social, and Governance) score (ranging from 0: poor ESG performance to 100: excellent ESG performance). ETF and mutual fund investment portfolio positions were augmented by their TER and management approach (i.e., active vs. passive). To establish a risk-free baseline, we queried the risk-free market rate (i.e., 10-year US treasury bill) from the Federal Reserve Bank of St. Louis online database FRED [70]. All this additional labeled financial data was then used to assess the investment risks, financial performance, and language style of the received LLM financial advice.

Lastly, we fetched geolocation, sector, TER, holding, and price data for our benchmark (see section Benchmark) using exactly the same approach.

## Financial performance analysis

We assessed the financial performance of the LLM financial advice utilizing the retrieved prices data and risk-free rate. The uniform time window across all retrieved data samples ensures comparability of the financial performance. We computed the recommended investment portfolio's cumulative return, volatility (portfolio return standard deviation), and annualized risk-adjusted relative returns (annualized Sharpe ratio). Portfolio positions lacking a ticker and price data (e.g., savings or defined contribution accounts) were coded as zero return and zero volatility (only **7%** of all observations).

## Language style analysis

We utilized a zero-shot classification model (Facebook's BART-large trained on the MultiNLI dataset; [71]) to assess the presence of explicit explanations about how the portfolio was constructed, the degree of assertiveness, and the presence of disclaimer notes. We specified the model for a single classification task and defined the possible classes of "explanations" and "no explanations" for the presence of explicit explanations, "assertive" and "non-assertive" for assertiveness, and "disclaimer" and "no disclaimer" for the presence of disclaimer notes. We retrieved the probabilities for each characteristic.

## ETF benchmark

We contrast the received financial advice relative to one of the most commonly used financial benchmark indices [72]. Specifically, we contrast all recommendations received by each LLM relative to the Vanguard Total World Stock Index Fund (VT) ETF. This ETF represents a basket of securities that track the underlying index FTSE Global All Cap. Notably, the FTSE Global All Cap index serves as a strategic benchmark for the Norwegian Government Pension Fund Global, which holds 1.5% of the world's listed companies [73]. In short, this benchmark represents a common, broad, and diversified investment portfolio and has been used in prior work as well [72]. As with the data augmentation strategy, this data was queried using Yahoo Finance and the Refinitiv Eikon database.

## Statistical analysis procedures

*Investment risks*. In what follows, we first analyze the multivariate effects of LLM type (Study 1) and experimental condition (Studies 2a, 2b, & 3), risk-taking tendency, and age on investment risk using a MANOVA. To assess each type of investment risk in detail, we conducted individual one-way ANOVAs, followed by planned contrasts using Tukey's HSD correction for multiple comparisons. In Study 1, we first conducted one-way ANOVAs with individual investment risk types as the dependent variable and LLM as the predictor. We then performed t-tests comparing each LLM to the benchmark

investment risk value. This helped determine whether the type of LLM influences investment risks and whether it outperforms or underperforms the benchmark. Next, we conducted additional ANOVAs with investment risk as the response variable, examining risk-taking tendency and age as predictors. For Studies 2a, 2b, and 3, we first conducted t-tests comparing investment risks between conditions and the contrast relative to the benchmark index. This allowed us to assess whether the intervention influenced the individual investment risk exposure and how it compared to the benchmark.

Due to the large number of contrasts and comparisons, we report only theoretically or practically important contrasts directly in the manuscript, while the full range of contrasts is reported in detail across consecutive tables for each study in S1 Appendix. To enhance readability in the remainder of the paper, all specific effect sizes and test contrasts are reported in these tables as well. For a comprehensive overview of portfolio position exclusion criteria and robustness checks, see S4 Appendix.

*Financial Performance Analysis.* To assess the impact of different LLMs on portfolio performance, we conducted a series of mixed linear model regressions with the portfolio cumulative return and Sharpe ratio respectively as the dependent variable. This approach was chosen to account for both fixed effects (i.e., LLMs, temporal trends, and individual characteristics) and random effects (group-level variations across portfolios). The fixed effects included the LLM (ChatGPT, Copilot, and Gemini) to assess whether specific models influence the return or Sharpe ratio. Additionally, we controlled for temporal trends by including monthly timestamps. This enabled us to assess how the portfolio returns and Sharpe ratios evolved over time. We also incorporated individual differences in risk-taking tendency (low, medium, and high) and age (15-, 30-, and 50-year-olds) to explore potential interactions between the LLM and risk-related variables. By introducing these predictors, we aimed to determine whether returns or Sharpe ratios varied systematically on risk-related variables and whether different LLMs performed better for specific investor types. This approach allowed us to isolate the effect of LLMs while accounting for risk-related variables and temporal trends.

*Language style analysis.* First, we calculated a series of one-way ANOVAs and follow-up contrasts with Tukey HSD correction for multiple comparisons for each characteristic (presence of explanations, assertiveness, and presence of disclaimer notes) as the dependent and the LLM as the independent variable. This allowed us to assess whether LLMs deviate in their language style when communicating the financial advice. Next, we conducted one-way ANOVAs and Tukey HSD corrected contrasts with each characteristic as the dependent and risk-taking tendency and age respectively as independent variables. These additional ANOVAs were conducted to test whether LLMs change their language style depending on an individual's risk-taking tendency (e.g., very assertive language for high risk-taking individuals) and age.

## Results

### LLMs increase all five investment risk types for private investors

In Study 1 we assessed the type of financial advice that private investors receive as a function of their appetite for risk (self-stated risk-taking tendency and investor age) from the most widely used LLMs (Open AI's ChatGPT, Google's Gemini, Microsoft's Copilot). Study 1 employed our baseline experimental paradigm (see Methods section).

A MANOVA with the investment risk types (geographical cluster, sector cluster, trend chasing, active investment allocation, and total expense risk) as the response and the LLM (ChatGPT/ Gemini/ Copilot), risk-taking tendency (low/ medium/ high), and age (15/ 30/ 50) as predictors showed significant effects of LLM type ($\Lambda_{Wilks} = .7$, $F(10, 464) = 9.20$, $p < .001$), risk-taking tendency ($\Lambda_{Wilks} = .19$, $F(10, 464) = 60.41$, $p < .001$), and age ($\Lambda_{Wilks} = .87$, $F(5, 232) = 6.81$, $p < .001$). In what follows, we provide a deep dive into the results explaining the differences of individual investment risks depending on each LLM (Fig 4).

*Geographical & sector cluster risk.* Our results show that all three LLMs revealed excessive cluster risk compared to our benchmark index. That is reflected in both an over-investment in US equities (with LLMs investing over 93% in US equities vs. 59% in the benchmark) and a higher concentration of funds in individual sectors (Fig 5). An additional concentration analysis showed that LLMs invested on average over 5% and 8% more in the already predominant technology

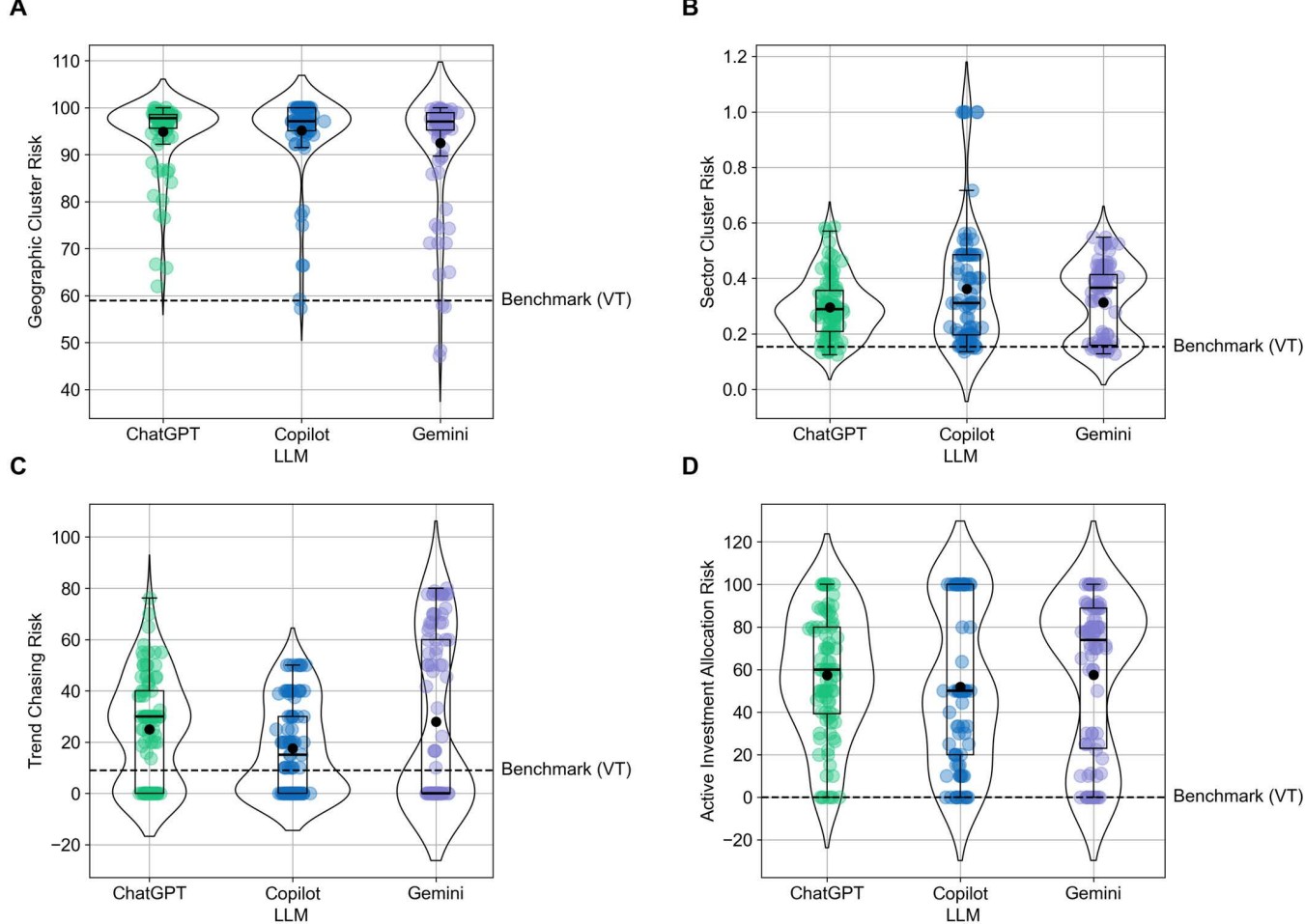

**Fig 4. LLM financial advice increases financial investment portfolio risks (Study 1).** (A) We find above benchmark geographical cluster risk (N = 269), (B) sector cluster risk (N = 269), (C) trend chasing risk (N = 270), and (D) active investment risk (N = 270). The violin plots and boxplots represent the shape of the distribution of the respective investment risk by LLM. The black dot represents the mean and the colored dots the value of individual portfolio samples. The dashed line represents the value of the benchmark.

(t(268) = 4.73, *p* < .001) and consumer cyclical sectors (t(268) = 8.23, *p* < .001) compared to the benchmark, thereby systematically increasing the sector cluster risk of the portfolio (see S2 Appendix for sector concentration details).

*Trend chasing risk.* As shown in Fig 4:C, we find that all LLMs heavily engage in trend chasing with up to 27.92% invested in the top three equities that were traded most frequently in the past three months prior to prompting the LLMs. Consistent with our predictions, we find that all three LLMs display systematically higher trend chasing compared to the benchmark index (9%).

*Active investment allocation risk.* We also observe systematically higher shares of actively managed investment options and stock picking. Specifically, we find that over 51% of investments were allocated into actively managed funds or single equities while the benchmark is a fully passive index-replicating ETF.

*Total expense risk.* Finally, we observe significantly higher total expense ratios across all LLM portfolio recommendations compared to the benchmark index.

Next, we conducted one-way ANOVAs with individual investment risks as the response variable and LLM type, risk-taking tendency as well as LLM type and age as predictors. Follow-up contrasts with Tukey's HSD correction revealed a

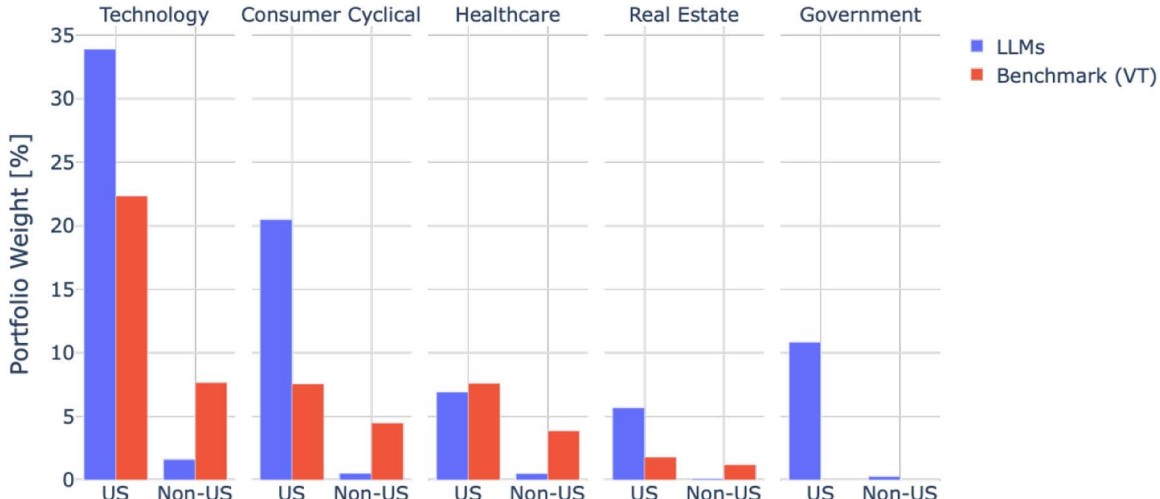

**Fig 5. LLMs show higher cluster risks compared to the benchmark.** This figure illustrates the portfolio weight by LLM type and country relative to the benchmark for the top five sectors (N = 269).

consistently higher active investment allocation risk for individuals with high (vs. low and medium) risk-taking tendencies. ChatGPT and Gemini even increased this risk for those with medium (vs. low) risk-taking tendencies. Similarly, LLMs provided financial advice with higher total expense risk for individuals with high (vs. low and medium) risk-taking tendencies. Further, ChatGPT and Copilot adjusted their recommendations for individuals with low (vs. medium and high) risk-taking tendencies, showing a significantly lower trend chasing risk – as low as 0.67% compared to 38.18% for those with high risk-taking tendencies. Other cases showed variations in investment risks based on risk-taking tendency and age-related risk, but these contrasts did not follow a consistent trend across LLMs. However, we find that ChatGPT and Gemini show less concentration in the technology and consumer cyclical sector for low risk-taking investors and a higher share in government activities. Copilot even concentrates over 46% of funds in government activities (i.e. bond ETFs) for low (vs. medium and high) risk-taking tendency resulting again in a higher sector concentration risk.

Taken together, Study 1 provides initial evidence that LLM investment advice induces a significantly enhanced investment portfolio risk across all five types of portfolio risks (geographical cluster risk, sector cluster risk, trend chasing risk, active investment allocation risk, and total expense risk). Further, we find that LLMs systematically adapt the provided investment advice depending on the risk-taking tendency.

*Financial performance analysis.* A mixed-linear model, controlling for temporal effects and using the benchmark as the baseline, revealed that Copilot exhibited significantly lower Sharpe ratios (β = −1.14, *p* < .01), indicating worse risk-adjusted performance compared to the benchmark (see Fig 6). In contrast, ChatGPT (β = −.28, *p* = .52) and Gemini (β = −.12, *p* = .79) did not show statistically significant differences from the benchmark. Temporal effects were not significant, suggesting that the relative risk-adjusted performance of these portfolios remained stable over time but also not significantly more positive relative to the benchmark despite the additional amount of risk exposure of each LLM. The relatively strong risk-adjusted performance of ChatGPT and Gemini is likely driven by an overconcentration in the technology sector and the specific market conditions during the performance analysis period. Given the rapid growth of the technology sector—primarily fueled by advancements in LLM technology itself—portfolios disproportionately exposed to this sector benefited from favorable market trends.

However, it is crucial to highlight that risk-adjusted returns, such as the Sharpe ratio, only adjust for the level of risk reflected in return volatility. This measure does not fully capture the broader spectrum of investment risks depending

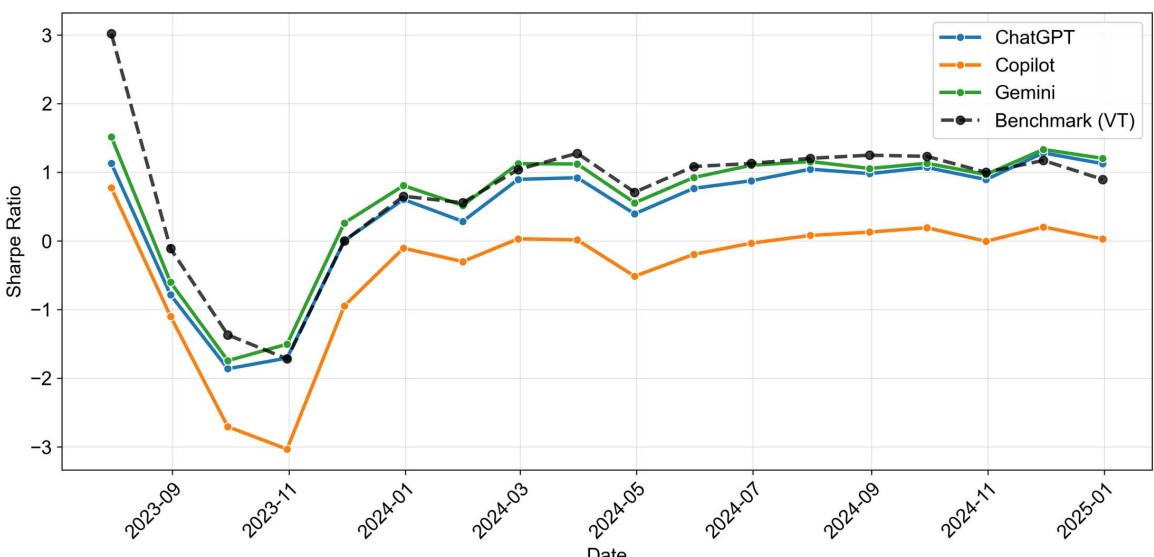

**Fig 6. Risk-adjusted returns relative to benchmark.** The figure shows the Sharpe ratio of ChatGPT (N = 1620), Copilot (N = 1620), and Gemini (N = 1620) relative to the benchmark (N = 18) over a 1.5-year period following the initial data collection.

on other market conditions. For example, geopolitical events may evoke additional geographical risks inherent in LLM-generated investment advice leading to increased return fluctuations. Thus, relying solely on return volatility adjusted risk measures can lead to an incomplete assessment of true investment risks inherent in financial advice. Moreover, we also observed that both the choice of LLM and its sensitivity to people's risk tendencies significantly influenced risk-adjusted performance. Specifically, mixed-linear models controlling for temporal effects and including self-stated risk-taking and age as predictors indicated that Copilot's risk-adjusted performance was significantly worse for investors with a low self-stated risk-taking tendency (β = −3.13, $p < .001$), as well as for 30-year-olds (β = −1.57, $p < .001$) and 50-year-olds (β = −1.12, $p = .01$) compared to the benchmark.

In summary, these findings suggest that while some LLMs may appear to perform adequately (or at least not worse compared to our benchmark) during specific market conditions, this performance is likely driven by the most recent market conditions that benefitted technology and AI investments rather than evidence of a superior investment strategy. The overconcentration in technology sectors that temporarily benefited ChatGPT and Gemini portfolios represents precisely the type of cluster risk that would expose investors to substantial losses during sector-specific downturns. Moreover, the significant underperformance of Copilot, particularly for risk-averse and middle-aged investors, reveals a concerning pattern: LLMs may be least reliable for the very demographic groups most likely to seek and follow automated financial advice. Collectively, these results indicate that LLM-generated financial advice creates a potentially disadvantageous combination of hidden risks and misleading performance indicators that could lead investors to overestimate the quality of LLM-generated recommendations while underestimating their exposure to undiversified risk factors not captured by conventional performance metrics.

*Language style analysis.* Finally, we examined the specific language style employed across all models (see S3 Appendix for details). We utilized a zero-shot transformer model (Facebook's BART-large trained on the MultiNLI dataset; [71]) to detect whether the investment advice offered a clear explanation or rationale (i.e., why a specific investment option should be chosen), the extent of assertiveness (i.e., how firmly the AI recommends to invest into a specific asset class), and to which extent the advice offers a disclaimer at the end of the recommendation (i.e., stating the potential risks involved with the investment). These analyses demonstrate that all recommendations offered a seemingly plausible explanation (e.g.,

"*Procter & Gamble is a stable company with a long history of paying dividends. Dividend stocks provide regular income and can grow over time*.")*, with medium to high assertiveness (e.g., "Considering your requirements, here's a diversified investment portfolio with a breakdown of how much you *should allocate* to each type of investment:"), and offering a seemingly trustworthy and "caring" disclaimer in the recommendation (e.g., "Remember, these are *just recommendations*, and it's crucial to do your own research or consult a financial advisor before making any investment decisions."). Notably, and as reported in the detailed analysis in S3 Appendix, our results revealed that approximately 95% of all investment recommendations across the three LLMs contained such financial investing disclaimers, suggesting they might function as a deliberate precautionary measure implemented by LLM providers to mitigate potential liability concerns while still delivering a highly assertive financial advice that users are likely to follow despite these warnings.

*Robustness checks.* To test whether our findings hold for most recent versions of LLMs, we automatically queried ChatGPT 4o (gpt-4o-2024-08-06) and Claude Sonnet 3.5 (claude-3–5-sonnet-20241022). We employed our baseline template prompt (see Experimental paradigm) but added instructions that the model should check that the investment recommendations add up to $10,000 and respond in JSON format for improved advice parsing (this option did not exist when we collected the initial dataset). We employed a 2 (LLM: GPT 4o/ Sonnet 3.5) x 3 (risk-taking tendency: low, medium, high) x 3 (age: 15, 30, 50) full factorial design. Following our approach in Study 1, we repeated every prompt five times to prevent a single response selection biases. We retrieved valid responses for all prompts (N = 90). Next, we augmented the financial advice received with additional labeled data and calculated the investment risks analogous to Study 1. Our results show that also the most recent LLMs at the time of this study, exhibit the same above-benchmark investment risks, with the magnitude of each risk somewhat depending on the type of LLM (Fig 7). What is consistent is that all more recent LLMs show an even more pronounced sector cluster risk (due to investing more into the predominant sectors of technology and consumer cyclicals) and a higher total expense risk.

**Narrow debiasing interventions only partially mitigate portfolio risks for private investors**

In Study 2a we assessed whether we can alter a single risk dimension (such as avoiding management fees) without changing the other types of risks. We used a two-cell experimental design (control prompt vs. debiasing intervention prompt) with a control prompt identical to Study 1 and a debiasing intervention prompt that explicitly requested no management fees ("*I don't want to pay any management fees.*"). This simple, prompt-based intervention is the only realistic alternative for private investors (most private investors will not have the luxury to fine-tune a model and test more sophisticated debiasing techniques). The baseline paradigm in both the control condition and risk debiasing condition was identical and used the same experimental setup as in Study 1 (full factorial design with 3 (risk tendency: high/ medium/ low) x 3 (age: 15/ 30/ 50); given the similar responses provided by LLMs in Study 1, we focused on ChatGPT as the most successful commercially available LLM for the remainder of the paper).

A MANOVA with the investment risks (geographical cluster risk, sector cluster risk, trend chasing risk, active investment allocation risk, and total expense risk) as the response and the prompting intervention (control vs. debiasing prompt), risk-taking tendency and age as predictors showed significant effects for the prompting intervention ($\Lambda_{Wilks}$ = .82, F(5, 154) = 6.6, $p < .001$), risk-taking tendency ($\Lambda_{Wilks}$ = .18, F(10, 308) = 41.68, $p < .001$), and investor age ($\Lambda_{Wilks}$ = .88, F(5, 154) = 4.31, $p < .01$).

Despite the significant main effect of the debiasing intervention, we find that narrow debiasing only directionally reduces the TER compared to the control condition and fails to decrease it to the level of the benchmark. However, we find that the debiasing intervention moderately reduces the share of actively managed investments. Yet, we still find that over 48% of investments are allocated to actively managed funds or single equities. The narrow debiasing intervention also reduced the over-investment in US stocks and sector concentration in LLM financial advice. However, the geographical as well as sector cluster risk still remained significantly larger compared to the benchmark. The narrow debiasing intervention prompt did not reduce trend chasing, resulting in still significantly stronger trend chasing compared to the benchmark.

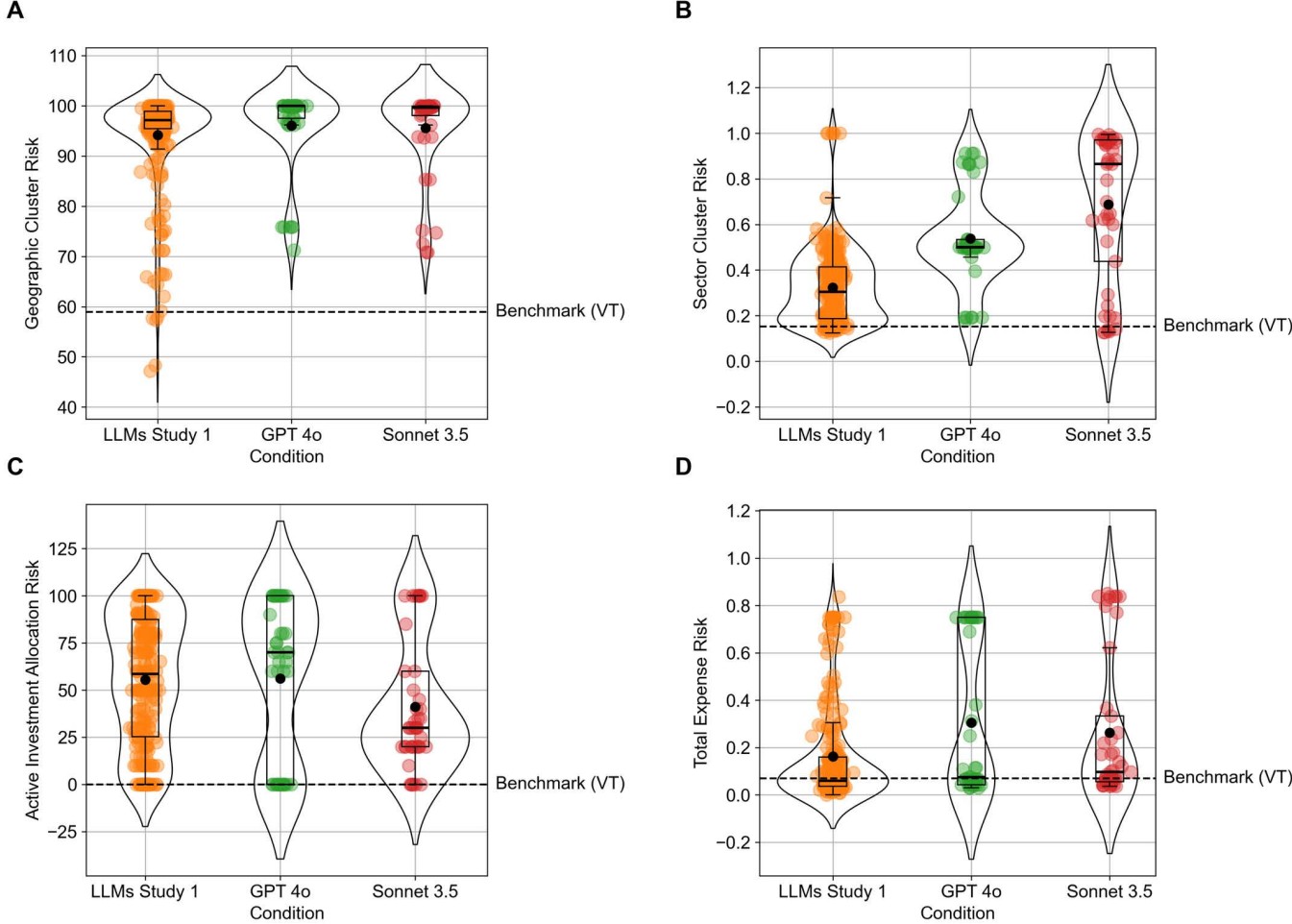

**Fig 7. More current LLMs show similar investment portfolio risks.** We find (A) above benchmark geographical cluster risk (N = 359), (B) sector cluster risk (N = 357), (C) trend chasing risk (N = 360), and (D) active investment risk (N = 270) also for current versions of LLMs. The violin plots and boxplots represent the shape of the distribution of the respective investment risk by LLM. The black dot represents the mean and the colored dots the value of individual portfolios. The dashed line represents the value of the benchmark.

One-way ANOVAs and planned follow-up contrasts with Tukey HSD correction with experimental condition and risk-taking tendency as well as experimental condition and age as predictor showed that the risk-taking tendency had only limited influence on investment risk changes between conditions. Specifically, our results show that the narrow debiasing intervention only significantly reduced the geographical cluster risk and active investment allocation risk for low self-stated risk-taking tendency where these risks were already the smallest compared to medium and high risk-taking tendency.

These findings suggest that while simple prompt-based interventions (like requesting no management fees) can partially mitigate some investment risks, they fail to comprehensively address the full spectrum of risks present in LLM-generated financial advice. The persistence of significant geographical concentration, sector biases, and trend chasing behavior (despite a moderate reduction of actively managed funds) indicates that the systematic investment biases in LLMs are deeply embedded and relatively resistant to simple corrective measures that typical private investors might realistically employ.

*Financial Performance Analysis.* A mixed-linear model, controlling for temporal effects and using the benchmark as a baseline revealed that the risk-adjusted performance of LLM investment advice in the control ($\beta = -.21$, $p = .19$) and debiasing intervention condition ($\beta = -.29$, $p = .08$) were not significantly different relative to the benchmark. Our results show that the risk-adjusted returns improved over time and were significantly lower in the first three months ($\beta_{08/2023} = -2.3$, $p = .02$; $\beta_{09/2023} = -3.43$, $p < .01$; $\beta_{10/2023} = -3.29$, $p < .01$). Furthermore, we find that especially for investors with a high ($\beta_{control} = -.41$, $p < .01$; $\beta_{debiased} = -.43$, $p < .01$) and low (vs. medium) risk-taking tendency ($\beta_{control} = -.46$, $p < .01$; $\beta_{debiased} = -.40$, $p < .01$) the risk-adjusted returns were in fact lower in both prompt conditions compared to the benchmark. Again, we also find that the risk-adjusted returns were especially lower in the short-term ($\beta_{08/2023} = -2.3$, $p = .01$; $\beta_{09/2023} = -3.43$, $p < .001$; $\beta_{10/2023} = -3.29$, $p < .001$). Finally, and similarly to Study 1, we find that 50-year-olds received investment advice with lower risk-adjusted returns ($\beta_{control} = -.34$, $p = .04$; $\beta_{debiased} = -.34$, $p = .04$) compared to the benchmark and that the risk-adjusted returns were especially reduced short-term ($\beta_{08/2023} = -2.3$, $p = .02$; $\beta_{09/2023} = -3.43$, $p < .001$; $\beta_{10/2023} = -3.29$, $p < .01$).

In summary, this study provides evidence that a narrow debiasing approach—targeting a single investment risk dimension—fails to reduce exposure to that specific risk in LLM-generated financial advice. Our findings suggest that private investors may only partially mitigate their investment risk exposure through simple prompt-based debiasing interventions. This is important as more advanced debiasing techniques, such as fine-tuning models, are not accessible to most private investors.

## Broader debiasing interventions are more effective in mitigating investment risks

In Study 2b we assessed to which extent a broader debiasing prompt may reduce the overall financial portfolio risk. Specifying multiple risks in the debiasing prompt, this prompting strategy may reinforce the risk reduction of every single risk alone as such broad debiasing interventions can often lead to effective debiasing of LLMs [74]. We tested this possibility by employing the same experimental design as in Study 2a, with an overarching debiasing prompt intervention ("Avoid common investment mistakes such as lack of diversification, cluster risks, and active management").

A MANOVA with the investment risks as the dependent and the prompting condition (control vs. debiasing intervention), risk-taking tendency, and age as the predictors showed a significant effect for the prompting condition ($\Lambda_{Wilks} = .62$, $F(5, 165) = 20.58$, $p < .001$), risk-taking tendency ($\Lambda_{Wilks} = .31$, $F(10, 330) = 25.84$, $p < .001$), while the effect of age was non-significant ($\Lambda_{Wilks} = .93$, $F(5, 165) = 2.14$, $p = .06$).

Follow-up contrasts revealed that this broader debiasing intervention (vs. the narrow debiasing intervention in Study 2a) was largely successful at reducing a range of investment risks (Fig 8). Specifically, we find that the debiasing prompt reduced the over-investment in US assets and sector concentration (while still remaining significantly greater compared to the benchmark however). The amount of assets allocated to US assets was reduced by over 10% but remained at a still high 80% level. Additional contrasts further show that the sector concentration decreased due to a significant 8% and 9% investment allocation decrease in the predominant technology and consumer cyclical sectors. These findings highlight that a more equal distribution across sectors is easier to achieve compared to an equal distribution across financial markets (i.e., arguably due to the strong presence of US-based equities). We find that the debiasing intervention also significantly reduced trend chasing to the extent that it no longer differs significantly from the benchmark. Simultaneously, we find that the debiasing intervention reduces the share of actively managed investment options by over 32%. Finally, the debiasing intervention also significantly decreased the TER by more than .06%. Nevertheless, TER remained notably higher compared to our benchmark.

One-way ANOVAs and planned follow-up contrasts with Tukey HSD correction with the prompting condition, risk-taking tendency and age as predictors revealed that the debiasing intervention was more effective for investors with a medium and high risk-taking tendency. We find that the debiasing intervention significantly decreased the sector cluster risk, trend chasing risk, and active investment allocation risk especially for investors with medium and high risk-taking tendency.

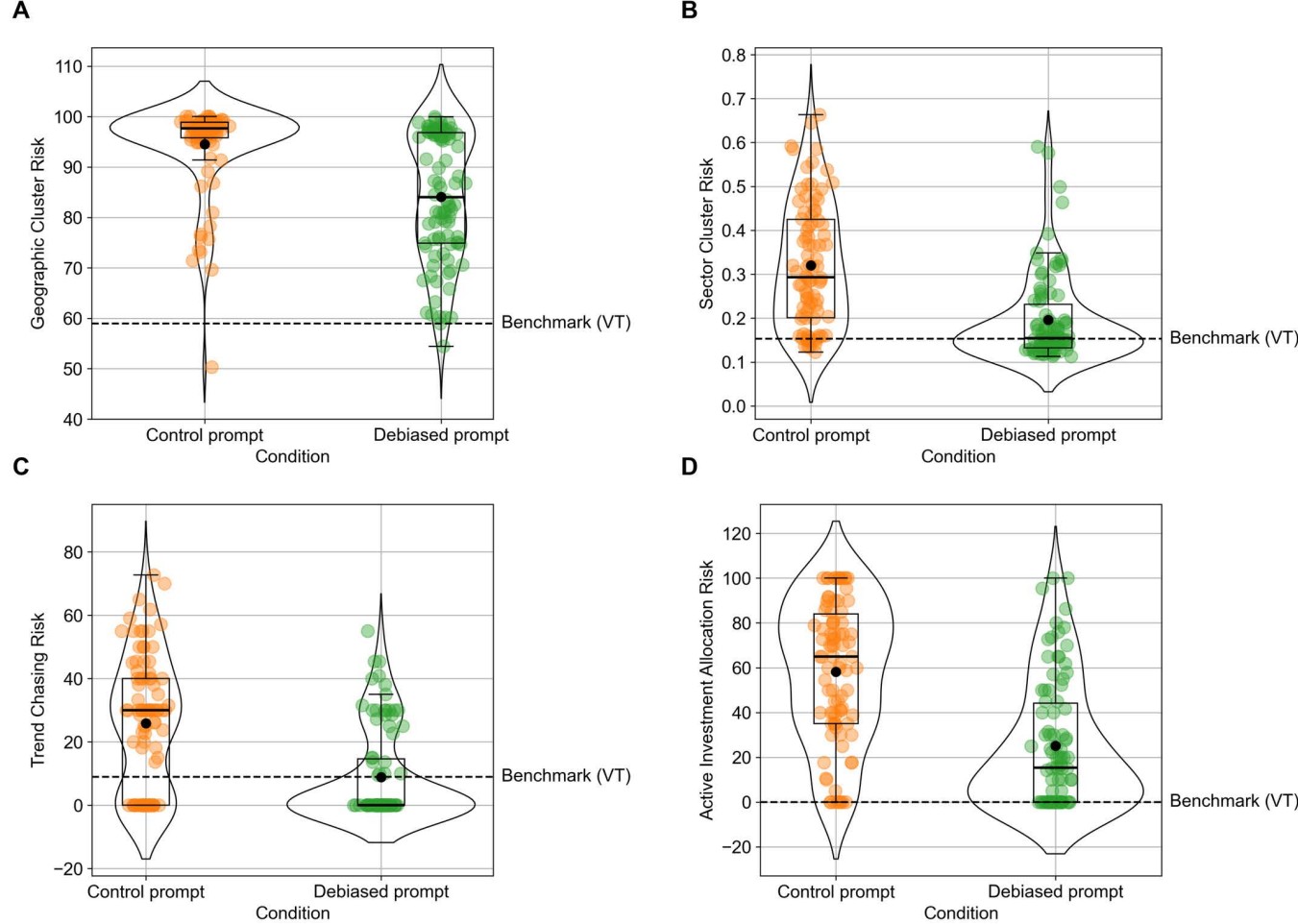

**Fig 8. Broad debiasing reduces the overall financial investment portfolio risks in LLM financial advice (Study 2b).** We find (A) above benchmark geographical cluster risk (N = 180), (B) sector cluster risk (N = 180), (D) active investment risk (N = 180), and (C) non-different trend chasing risk (N = 180) when using a broad debiasing intervention. The violin plots and boxplots represent the shape of the distribution of the respective characteristic by condition. The black dot represents the mean and the colored dots the value of individual portfolios. The dashed line represents the value of the benchmark.

Additionally, the debiasing intervention also decreased the total expense risk for high risk-taking tendency and the sector cluster risk, trend chasing risk, and active investment allocation risk across all age levels.

The collective insight is that broader debiasing interventions explicitly targeting multiple investment risks simultaneously can significantly reduce portfolio risks across most dimensions and age groups, although the geographic concentration remains somewhat elevated above benchmark levels. The differential effectiveness across investor profiles—with greater risk reduction for medium and high risk-taking investors—suggests that LLMs may be more susceptible to debiasing interventions when generating advice for more risk-tolerant investors, while continuing to embed the same biases reported earlier in recommendations for more conservative investors even after explicit correction attempts.

*Financial Performance Analysis.* Analogous to Study 2a, we performed multiple mixed-linear regression models. Again, we find no significant difference of the LLM investment recommendations provided in the control ($\beta = -.25$, $p = .11$) and the debiased condition ($\beta = -.29$, $p = .06$) relative to the benchmark in terms of risk-adjusted returns. We only find a moderate temporal trend indicating lower risk-adjusted performance especially with reduced short-term returns ($\beta_{08/2023} = -2.26$,

$p=.02$; $\beta_{09/2023}=-3.46$, $p<.001$; $\beta_{10/2023}=-3.43$, $p<.001$). We find that the risk-adjusted returns are worse for high ($\beta=-.40$, $p<.01$) and low risk-taking tendency ($\beta=-.48$ $p<.001$) in the control condition and for low risk-taking in the debiased condition ($\beta=-.37$ $p=.01$). Further, we find again that the risk-adjusted returns are lower than the benchmark for 50-year-olds in the control condition ($\beta_{control}=-.34$, $p=.03$) and for 30-year-olds in the debiased condition ($\beta_{debiased}=-.37$, $p=.02$). Also, in this model we find that the risk-adjusted returns were especially reduced short-term at the beginning of the performance measurement period ($\beta_{08/2023}=-2.26$, $p=.01$; $\beta_{09/2023}=-3.46$, $p<.001$; $\beta_{10/2023}=-3.43$, $p<.001$).

Thus, despite the successful risk reduction through broader debiasing interventions, both control and debiased LLM-generated portfolios still failed to outperform the benchmark on risk-adjusted returns. These results suggest that even when successful at mitigating specific portfolio risks, debiasing interventions do not translate into improved risk-adjusted performance, indicating that LLM-generated financial advice continues to carry inefficiencies that cannot be fully addressed through prompt engineering alone.

### LLMs effectively adapt financial advice to specific investment goals but induce new concentration issues

In Study 3 we tested whether incorporating an explicit investment goal in the prompt further aids in mitigating specific investment risks in LLMs investment advice. The study directly tests the ability of LLMs to adapt their investment advice based on contextual information and to personalize the investment advice to specific investor needs. Instead of directly asking the LLM to reduce the investment risks of a portfolio, a private investor may prompt a specific investment goal that would in turn translate into a decrease of specific investment risks merely based on the contextual information provided. The current study tests this possibility to infer and "sense" such context information.

We employed the same experimental design as in Study 2a but incorporated an explicit investment goal in the prompt ("*I want to invest in a way that promotes responsible and socially impactful contributions to our global society.*"). We anticipated that this goal would lead LLMs to offer financial advice with reduced geographical and sector concentration risks. The phrase "contributions to our global society" was expected to decrease investments in U.S. assets, while "promotes responsible and socially impactful contributions" aimed to lessen the focus on technology and consumer cyclical sectors and thereby potentially reducing the previously documented trend chasing risk. We also expected that incorporating such a goal would result in investment recommendations with higher ESG scores [75].

A MANOVA with the investment risks as response and the prompt condition (control vs. goal prompt), risk-taking tendency and age as predictors showed a significant effect of the prompt condition ($\Lambda_{Wilks}=.30$, F(5, 161) = 74.31, $p<.001$), risk-taking tendency ($\Lambda_{Wilks}=.51$, F(10, 322) = 12.67, $p<.001$), while showing only a marginal effect of age ($\Lambda_{Wilks}=.94$, F(5, 161) = 2.20, $p=.06$).

Follow-up contrasts revealed that investment risks showed a significant decrease in over-investments in US Stocks in the investment goal prompt compared to control, yet higher than our benchmark. However, we find that the introduction of a socially responsible investment goal did not lead to a significant decrease in sector concentration compared to the control prompt and remains still significantly higher than the benchmark index. As shown in Fig 9:A, not only remains this sector concentration very high, but is also substantially overleveraged towards utilities, due to shifting investments towards the energy sector (i.e., "green" energy). Next, we further examined the ESG score in the intervention (vs. control) condition. As summarized in Fig 9: B, specific investment goals were not effective in increasing the portfolio ESG rating of the portfolio. Specifically, we find that there is no significant portfolio ESG rating difference when incorporating a social responsibility goal prompt compared to the control prompt or benchmark. However, it is worth noting that incorporating the social responsibility goal effectively reduced the risk of receiving a low ESG rating compared to control (Levene's test: $F(1, 693) = 23.83$, $p<.001$). We also observe a significant decrease in trend chasing behavior in the LLM investment advice when incorporating a social responsibility investment goal. Adding the social responsibility investment goal however, increased the actively managed portfolio share by over 29% compared to the control condition. Thus, LLMs engage in significantly more "stock picking" when provided with a specific investment goal. Finally, we observe significantly higher total expense ratios when incorporating a social responsibility goal compared to our control condition and benchmark.

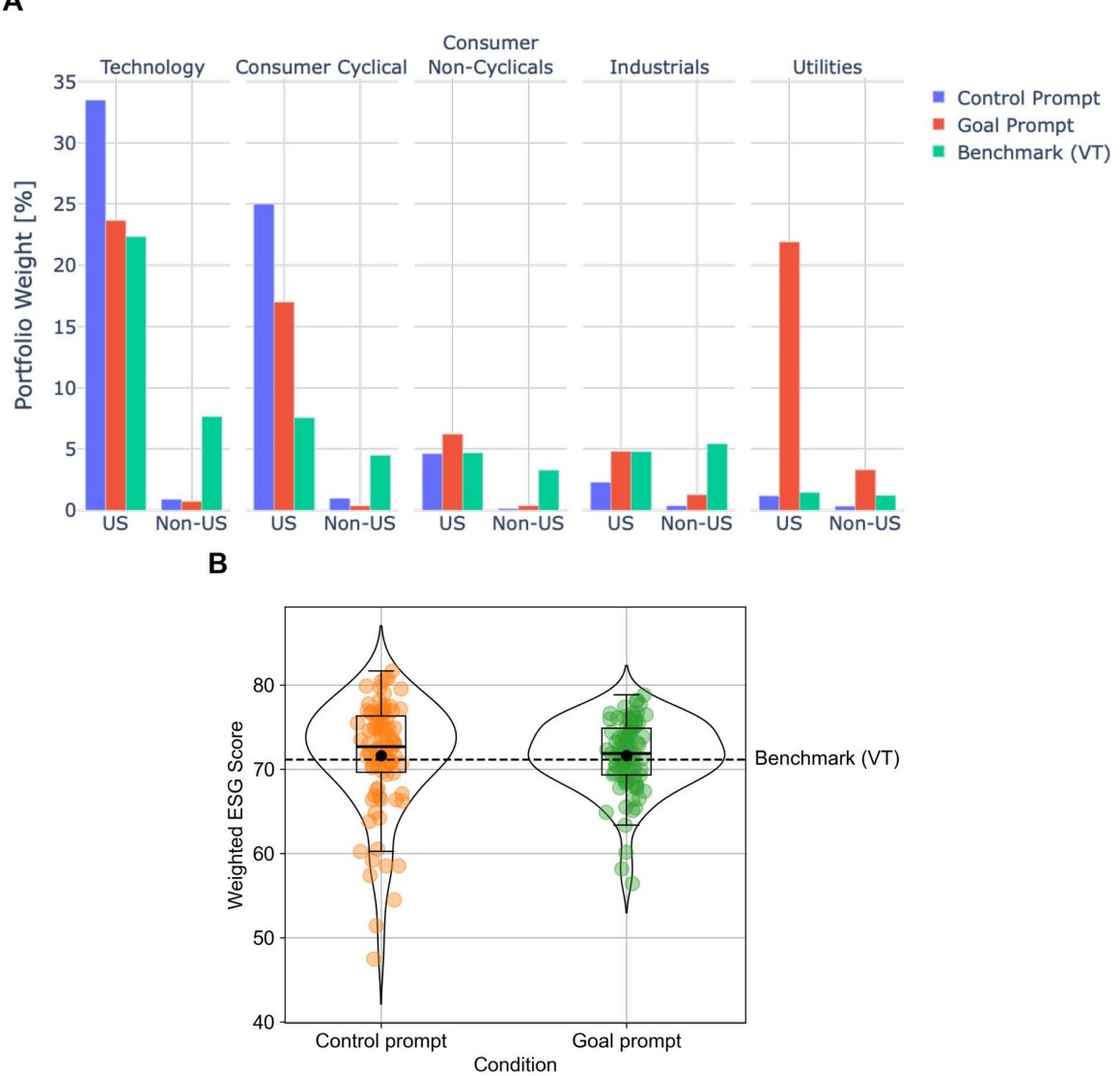

**Fig 9. Incorporating a social responsibility goal in the prompt causes overleverage in utilities (A) and decreases risk of low ESG rating (B) (Study 3).** (A) illustrates the portfolio weight by condition and country relative to the benchmark for the top five sectors (N = 180). In (B) (N = 180) the violin plots and boxplots represent the shape of the distribution of the ESG score by condition. The black dot represents the mean and the colored dots the value of individual portfolios. The dashed line represents the ESG score value of the benchmark.

One-way ANOVAs and planned follow-up contrasts with Tukey HSD correction revealed that incorporating the social responsibility goal in the prompt led to a significant increase in active investment allocation and total expense risk for participants with low and medium risk-taking tendencies. In contrast, trend chasing risk decreased for those with medium and high risk-taking tendencies under the same condition. Further, the ESG score only increased for high risk-taking tendency when including the socially responsible investment goal where the ESG score was significantly lower (vs. medium and high self-stated risk-taking tendency) in the control condition. The geographical cluster risk decreased only for 50-year-olds. All other contrasts did not yield any meaningful patterns.

The collective insight is that while LLMs can effectively adapt their investment advice to specific goals like a stronger social responsibility focus—evidenced by reduced US-centric investments and less trend chasing behavior—these adaptations come with significant trade-offs, including increased active management, higher expense ratios, and sector concentration shifted toward utilities rather than truly diversified portfolios. The inconsistent effectiveness across investor profiles further suggests that LLMs struggle to simultaneously optimize for both specific investment goals and fundamental portfolio construction principles, creating potentially new risk patterns when attempting to accommodate more personalized investing objectives.

*Financial Performance Analysis.* As in the preceding studies, we conducted a series of mixed-linear models to analyze portfolio performance. We find that investments recommended by LLMs in the control condition are not significantly different from the benchmark in terms of risk-adjusted returns ($\beta = -.25$, $p = .15$). In the debiased condition, the LLM investment advice even underperformed the benchmark significantly ($\beta = -.76$, $p < .001$). The temporal trend indicating worse risk-adjusted performance in the beginning holds also for this model ($\beta_{08/2023} = -2.62$, $p = .01$; $\beta_{09/2023} = -3.76$, $p < .001$; $\beta_{10/2023} = -3.78$, $p < .001$; $\beta_{10/2023} = -2.20$, $p = .04$). We find that this underperformance in the debiased condition stems from the underperformance across both high ($\beta = -1$, $p < .001$) and medium risk-taking tendency investors ($\beta = -1.02$, $p < .001$) and we also find a clear underperformance for low risk-taking tendency in the control condition ($\beta = -.48$, $p < .01$). While we find no significant risk-adjusted underperformance for any age in the control condition, we document a strong underperformance across all age groups in the debiased condition ($\beta_{15} = -.70$, $p < .001$; $\beta_{30} = -.75$, $p < .001$; $\beta_{50} = -.85$, $p < .001$).

Thus, we find overall that the introduction of socially responsible investment goals in LLM-generated advice led to significant underperformance compared to the benchmark, highlighting that when LLMs attempt to optimize for specific social impact objectives, they tend to sacrifice financial performance even more severely and lead to new, unwanted concentration issues (i.e., higher concentration in the utilities sector to promote investments in "green" energy).

## Discussion

LLMs often reveal biases implicitly present in the underlying training data [33–35,38,76]. Our findings suggest that these biases can result in LLM-generated financial advice that leads to elevated investment risks for private investors across five major risk dimensions (geographical cluster risk, sector cluster risk, trend chasing risk, active investment allocation risk, and total expense risk) and that a range of debiasing interventions only partially mitigated these risks. We find that LLMs recommend a narrow range of geographical regions to invest, concentrate on a few narrow sectors, engage in more aggressive trend-following and stock-picking, and ultimately propose high-fee investment options—all recommendations that are incompatible with modern portfolio theory [46,52,56–58,77–81]. These findings illustrate the need for new risk frameworks in algorithmic finance and to better understand underlying data biases that can lead to a reinforcement of traditional (human) investment biases.

The findings of this research document the inherent risks for private investors who already started to employ LLMs to receive financial advice [15], and we hope that the current findings promote a more critical assessment of the advice they receive. However, it is noteworthy to highlight that a critical assessment becomes more difficult when LLM-generated investment advice is communicated very convincingly (see S3 Appendix for details). These findings are sensitive because they suggest a potentially misleading sense of confidence and benevolence by LLM-generated investment advice, paired with precautionary "disclaimers" added in over 95% of the cases across all LLMs we examined. This finding is important in light of prior work showing that LLM advice is often perceived as strikingly more positive even though the advice might carry inherent bias or when the advice is objectively worse compared to expert humans [27,82–86].

The ability to blindly follow harmful LLM-generated investment recommendations presents a significant risk for individual investors, particularly those lacking potentially financial literacy or access to professional financial advisors. Future

research should investigate how personal financial goals could be better integrated into AI investment strategies while maintaining appropriate risk management. A fruitful direction for future work is also to further examine whether financial education (potentially even through the use of LLMs) can ultimately empower private investors to critically assess such LLM-generated investment recommendations. Future regulatory research may also focus on developing frameworks to protect consumers from the potential risks of such generative investment advice. This includes exploring effective monitoring mechanisms to prevent more large-scale, systemic risks across financial markets and establishing reporting requirements for firms that utilize generative investment advisory tools. As LLMs become more integrated into financial advisory services, regulators may need to establish clear guidelines for transparency, explainability, and accountability in such AI-driven financial advice, especially when used by private investors. For FinTech developers, our research highlights the importance of addressing inherent biases in LLM-generated financial advice and how to develop future systems that better align with modern portfolio theory principles and avoid reinforcing harmful investment biases documented in the current research. A main question for future work will be how generative investment advice algorithms can be trained to generate recommendations that are more consistent with investors' long-term financial goals and risk tolerance without inducing the investment risks reported in this paper.

As with any research, we also highlight at least two limitations of the current work. First, we evaluated LLMs at specific points in time, and these models continue to evolve very rapidly. While our robustness checks with the most recent models (as per January 2025) seem to be robust (see Robustness Checks at the end of Study 1), future generations of models might improve radically once foundation models might be explicitly debiased during the training stage. Second, our simulated investment portfolios were evaluated over a specific time-period that coincided with a strong technology sector performance. However, we believe that exactly due to these favorable market conditions for technology-heavy portfolios, our financial performance analysis reported in this paper represent a conservative estimate of the potential risks of LLM-based financial advice. In market environments where technology sectors underperform, the geographical and sector concentration risks we identified would likely lead to even worse investment outcomes, further amplifying the risk exposure of portfolios generated by LLMs.

In conclusion, the current work demonstrates that LLM-generated investment advice systematically increases portfolio risks across multiple dimensions, potentially exposing private investors to unwanted investment risks while creating a false sense of confidence through convincing language and reassuring disclaimers. These findings underscore the need for more cross-disciplinary collaboration among investors (both private and institutional investors), regulators, FinTech innovators, financial institutions, and researchers to develop more robust frameworks for evaluating and improving AI-generated financial advice. We hope the current research promotes such collaborations to proactively transform the financial advisory landscape to truly serve the interests of private investors through properly diversified, low-cost, and personalized financial advice.

## Supporting information

**S1 Appendix. Portfolio investment risk contrasts.**
(DOCX)

**S2 Appendix. Extended concentration analysis.**
(DOCX)

**S3 Appendix. LLM financial advice language style analysis.**
(DOCX)

**S4 Appendix. Exclusion criteria and robustness checks.**
(DOCX)

## Acknowledgments

We thank the reviewers for their thoughtful feedback and constructive suggestions that significantly improved this manuscript. We thank the participants at the Wharton Generative AI conference, TUM Generative AI conference, the Association for Consumer Research conference, and the lab meetings at the University of St.Gallen for their feedback. We would also like to thank Stefano Puntoni and Noah Castelo for their feedback on earlier versions of this paper. Special thanks to Patrik Schär from Selma Finance for providing an industry perspective and practical comments that helped strengthen our research.

## Author contributions

**Conceptualization:** Philipp Winder, Christian Hildebrand, Jochen Hartmann.

**Data curation:** Christian Hildebrand.

**Formal analysis:** Philipp Winder, Christian Hildebrand.

**Investigation:** Philipp Winder, Christian Hildebrand.

**Methodology:** Philipp Winder.

**Project administration:** Philipp Winder.

**Supervision:** Christian Hildebrand, Jochen Hartmann.

**Validation:** Christian Hildebrand, Jochen Hartmann.

**Visualization:** Philipp Winder.

**Writing – original draft:** Philipp Winder, Christian Hildebrand, Jochen Hartmann.

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
