## [Decision Letter · Decision Letter 0]

Dear Dr. Winder,

Thank you for submitting your manuscript to PLOS ONE. After careful consideration, we feel that it has merit but does not fully meet PLOS ONE’s publication criteria as it currently stands. Therefore, we invite you to submit a revised version of the manuscript that addresses the points raised during the review process.

We look forward to receiving your revised manuscript.

Kind regards,

Peter Gordon Roetzel

Academic Editor

PLOS ONE

Journal Requirements:

2. Please include captions for your Supporting Information files at the end of your manuscript, and update any in-text citations to match accordingly. Please see our Supporting Information guidelines for more information: http://journals.plos.org/plosone/s/supporting-information .

Reviewers' comments:

Reviewer's Responses to Questions

**Comments to the Author**

1. Is the manuscript technically sound, and do the data support the conclusions?

Reviewer #1: Yes

Reviewer #2: Yes

2. Has the statistical analysis been performed appropriately and rigorously?

Reviewer #1: Yes

Reviewer #2: Yes

3. Have the authors made all data underlying the findings in their manuscript fully available?

Reviewer #1: Yes

Reviewer #2: Yes

4. Is the manuscript presented in an intelligible fashion and written in standard English?

Reviewer #1: Yes

Reviewer #2: Yes

Reviewer #1: !! - see attached document for better readability.

--- ---

1. Foreword

This research investigates the advice given by generative AI (henceforth, GenAI) in investment strategies for private investors. Three GenAIs representing the most common GenAIs used by the general public were used. The prompts to generate the AI advice included variables such as the investor's age and willingness to take risks in their investments. Contrasting the investment advice offered by a trusted investment source, it was found that the advice given by GenAIs showed typical biases in finance and investment. More specifically, GenAI advice tended to propose overly risky investments, i.e. investments with little geographical diversity (i.e. mainly in the US) and sector diversity (generally in one sector at a time), investments linked to recent trends (last 3 months with the risk of buying high and selling low), active investment allocations (risk of high transaction costs and underperformance) and investments with high management and operating costs measured by the Total Expense Ratio (or TER; where preference should be given to low TERs, i.e. lower fees). It was also observed that, despite additional statements in prompts calling for varied investments (Study 2a; narrow debiasing), investments avoiding typical investment biases such as lack of diversity, clusters or active management (Study 2b; broad debiasing) or specific investment goals (Study 3), these statements only mitigated these risks without cancelling them out.

This research addresses a crucial theme regarding the expansion and increasing accessibility of generative AI to the general public. Indeed, it is clear that the performance of these tools is sufficiently convincing for an untrained private investor to be attracted to using them for monetary investments. Nevertheless, as the research rightly points out, the lay investor can risk financial losses without training or a critical eye regarding these pieces of advice. The IT resources used for testing these effects are of outstanding quality. They are readily reproducible (obviously, within the limits that GenAIs never generate strictly identical outputs) and adaptable for further research using the same tools. While it is a criterion for publication in PONE, the availability of computing and analytical resources is a significant advantage for research in this area and for furthering this research.

While the quality of the tools and methods used to achieve these results is indisputable, I have significant concerns regarding the presentation format, which needs substantial improvements. The article’s deviation from the typical structure of experimental studies may hinder comprehension. These formatting issues do little to promote its qualities. The theoretical framework is too brief. The methodology lacks several essential components, such as the specifics of the analysis performed. The discussion does not address the relationship between this research and prior studies, as well as its limitations and further research directions [after second reading: I realise that all this is in the Supplementary Information and deserves to be in the article’s main body]. 

--- ---

2a. Introduction – General comments

To begin with, the introduction is too brief for a topic as crucial as the role of AI and the risks associated with trusting potentially risky financial investment advice. The research establishes part of the territory in the first paragraph (26 – 45) by explaining the importance of the study: three articles highlight the rising application of GenAI in financial investments by private investors (i.e., 1–3), while only one study provides concrete statistics related to the population in the UK (i.e., 4). Insufficient studies are presented, especially since they address the issues arising from the increasing reliance on GenAI. Moreover, while the literature may, let’s suppose, be scarce when it comes to the use of GenAI for financial advice, there might be relevant studies in other domains where overreliance can be detrimental and problematic. For instance, overreliance can have adverse effects in fields such as professional relationships, academic advice, etc.

The second paragraph (47 – 58) sets out in more detail how GenAIs are prone to various biases, with several scientific articles supporting the point. However, the second paragraph ends by already presenting the research hypotheses and predictions. Indeed, the importance of research is established, and we have learned that more and more private investors are using GenAI advice and that GenAIs have significant biases. However, the paragraph does not establish the niche: we know neither whether there are counter-arguments to the use of GenAI for financial investments (i.e. that there may be benefits to its use) nor whether there is a gap in this issue, nor does it present questions that current research has not yet been able to answer. This paper does not show any limitations of previous research, how it can be developed further, or what it has missed. All these points are aimed at occupying the niche and giving a solution. Finally, the third paragraph (59 – 67) presents the results obtained right before detailing the method, somewhat echoing what was stated in the abstract. A brief overview of the ‘present study’ would have been appreciated instead.

Finally, the introduction uses key terms yet fails to clarify or define them in detail. While finance experts may easily understand these terms as they have the necessary background, the scope of this research goes beyond this strict area. It should be accessible to individuals from other fields and, ideally, the general public. This research highlights the dangers of non-experts’ over-reliance on GenAI, non-experts who are not aware of those risks. Unfortunately, this article’s target audience may struggle to appreciate its implications fully. However, I must acknowledge that these terms are in Table 1, which contains nearly sufficient information to address and understand them. This table, only found in the middle of the Method section, summarises the measures, their definitions, their risks, and the calculations used for their accurate measurement. This table summarises definitions that should have been extensively defined beforehand, i.e., at the beginning of the theoretical background.

---

2b. Introduction – Small comments

l.48 Even if it is intuitive, is there any source that attests to this claim?

l.43 – 45 These terms should have been defined beforehand to thoroughly understand their implications for the research. There is a risk of back-and-forth-reading between paragraphs.

--- ---

3a. Methods – General comments

In the beginning, I thought the mentioned factors of the risk tendency, the age, and the GenAI were conditions whose levels would be compared for each study in a comprehensive analysis, as it would have been done in a repeated-measures ANOVA or MANOVA (or linear mixed models with iterations as random effects). As mentioned in lines 72 to 76, we have the impression that a comparison between older and younger people would be made in the same manner as a comparison between the different levels of risk-taking (where higher risk-taking could supposedly lead to more significant biases), or even which GenAI would cause the riskier investments.

No comparison between these levels is made in the Results section of the primary document. We then understand in the Results section that these variations in the prompts are used to give a more representative sample of prompts possibly used by private investors and that these are repeated over ten iterations each, i.e., 27 cross-conditions by 10 iterations for 270 occurrences in total. These iterations are presented as a factor in the experimental design: if it were the case, the levels of this factor would need to undergo pairwise comparisons (e.g., iteration 1 to iteration 2, iteration 1 to iteration 3, etc.). I would recommend presenting the interactions strictly as iterations of the three previous variables, not as a factor per se.

Nevertheless, these comparisons between risk-tendency, age, and GenAI levels are carried out in the Supplementary Information (SI) but should be presented in the manuscript. Indeed, the SI has enormous potential for analyses that deserve to be presented in the manuscript itself. Moreover, SI should be mentioned less in the manuscript. The manuscript should stand on its own as the article, so we do not have to read/refer many times to the SI.

Finally, as we reach the conclusion of the section, there is no presentation of the analyses carried out (it seems there are several, in fact): which tests are used? how many? each measure one by one (several ANOVAs at the risk of increasing likelihood of alpha error)? a single and comprehensive analysis (e.g., MANOVA with 5 dependent variables)? which comparisons are carried out (Tukey HSD, LSD, Bonferroni,…)?

---

3b. Methods – Small comments

l.74-76 It is not clear who should invest in less risky assets, the younger or the older investors?

l.104 TER refers to Total Expense Ratio, but the abbreviation was not introduced earlier. The financial term ‘Total Expense Ratio’ is not defined earlier (unless I am mistaken).

l.109 – 110 This builds on my earlier point regarding content that might not be understandable to those unfamiliar with the field. The abbreviation ‘FRED’ is neither introduced nor explained, leaving me – and/or others – unable to grasp the sentence fully.

--- ---

4a. Results – General comments

In general, the studies’ results are well presented. We know where the comparisons are significant and where GenAI is a problem when compared to the benchmark. The results are also interesting: even with the investment disclaimer avoiding bias (Study 2a, 2b, 3), some of them still stand out. However, I recommend compiling all these descriptive statistics into a summary table for easier readability and to allow you more room to elaborate on each result, such as which sectors or geographical areas are over- or under-represented.

I found the paragraph (168 – 177) on portfolio returns intriguing, not merely ‘ancillary,’ as it deserves more attention. This raises an interesting question: despite the biases noted in the literature, do investments made by GenAI still yield returns when compared to those made with trustworthy devices? What about each prompt? Despite the identified biases, could GenAI ultimately have merit? Perhaps only over a span of 3 or 6 months, and less so over 1 or 2 years (to the extent we can project that far back from when the AI outputs were generated)?

---

4b. Results – Small comments

l.145 Which sectors specifically?

l.163 I am surprised that the benchmark does not suggest any actively managed funds or single equities.

l.276

l.275 – 280

l.281 – 287 The ESG score is not mentioned anywhere before, either in the method or in the foreword to the results, nor is its abbreviation presented (i.e., Environmental, Social and Governance score). For a non-expert, it is impossible to guess that ESG represents a score ranging from 0 to 100, where the higher the score, the better it is, and represents high environmental and ethical performance.

--- ---

5a. Discussion – General comments

As for the introduction, the discussion is too brief and could benefit from further elaboration. The discussion does not mention the links with existing research that would have been presented in the introduction, what answers it provides, what other research it contradicts or agrees with, or the hypotheses it accepts or rejects. Additionally, the discussion does not mention the limitations of the present methodology; variables deliberately left out that could be explored in future studies. Regarding future studies, what new directions and questions could it answer? On this last point, the SI (section 9; p. 22) does suggest some questions that could be further elaborated in the main article.

While it is observed that GenAI carry significant biases, the research does not enough emphasise the importance and presence of the disclaimers, specifically the caution against over-reliance on the advice given and the necessity to consult a professional in the field. Notably, the disclaimers in section 5 (p. 17) of the SI are over 75% of the output content, which is substantial (which is the result of numerous reports and precautions taken by the companies behind the GenAIs).

---

5b. Discussion – Small comments

l.310 I have some reservations about the term ‘critical guide’. While the present article highlights the biases inherent in GenAI, I agree that it shows in what GenAI can be an unreliable adviser. However, it does not establish a checklist or precise safeguard that investors must follow to use the advice of a GenAI ‘in a fully informed manner’. It acknowledge the risks, but does not offer any proper guidance.

--- ---

6. Concluding comments

My comments might seem a bit critical, but I genuinely believe that the methodology is sound, valid and reliable. As I said, the format conceals the quality of the substance of this research. I was pleasantly impressed by how the methodology allows for reliable comparisons between GenAI and benchmark advice. Furthermore, the calculations for assessing risk levels are reliable and objective, and the sources (e.g., the benchmark itself) are of high quality. Table 1 is excellent and provides a good overview of the key measures of this research.

Reviewer #2: This research paper ( Biased Echoes: Generative AI Models Reinforce Investment Biases and Increase

Portfolio Risks of Private Investors) It's a pretty nice , and thanks for the efforts been made on writing this paper. But in order to be published, there are some modifications need to be done. Frist, (in Broader debiasing interventions are more effective in mitigating portfolio risks) the authors need to explain more in study2 regarding the risks that accurate. Also, in (Discussion) section if the authors illustrate more on the studies that been doing.

over all the article is very good, just to make these modifications, the paper will be ready to be published.

**Do you want your identity to be public for this peer review?** For information about this choice, including consent withdrawal, please see our Privacy Policy

Reviewer #1: **Yes: ** Daniel de Oliveira Fernandes

Reviewer #2: **Yes: ** Nawaf Abdualaziz Almolhis

---

## [Author Response · Author response to Decision Letter 1]

9 Mar 2025

Remark: Please see the attached document for better readability!

Revision notes for:

“Biased echoes: Large language models reinforce investment biases and increase portfolio risks of private investors” (PONE-D-24-45837)

We appreciate the review team’s thoughtful and constructive comments on our initial submission. In revising the manuscript, we closely followed the guidance provided by each member of the review team. We are confident that we have addressed all major issues raised by all members of the review team, producing what we believe is a much stronger paper. In what follows, we respond to the specific comments provided by each member of the review team (original comments in italics, followed by our responses).

----

Responses to Reviewer’s 1 comments

This research addresses a crucial theme regarding the expansion and increasing accessibility of generative AI to the general public. Indeed, it is clear that the performance of these tools is sufficiently convincing for an untrained private investor to be attracted to using them for monetary investments. Nevertheless, as the research rightly points out, the lay investor can risk financial losses without training or a critical eye regarding these pieces of advice. The IT resources used for testing these effects are of outstanding quality. They are readily reproducible (obviously, within the limits that GenAIs never generate strictly identical outputs) and adaptable for further research using the same tools. While it is a criterion for publication in PONE, the availability of computing and analytical resources is a significant advantage for research in this area and for furthering this research.

>Thank you for your favorable assessment of the initial version of this manuscript! We greatly appreciate your many thoughtful suggestions, which we followed closely in revising the paper. Following your suggestions led to a much stronger paper both conceptually and empirically. Thank you!

_

While the quality of the tools and methods used to achieve these results is indisputable, I have significant concerns regarding the presentation format, which needs substantial improvements. The article’s deviation from the typical structure of experimental studies may hinder comprehension. These formatting issues do little to promote its qualities. The theoretical framework is too brief. The methodology lacks several essential components, such as the specifics of the analysis performed. The discussion does not address the relationship between this research and prior studies, as well as its limitations and further research directions [after second reading: I realise that all this is in the Supplementary Information and deserves to be in the article’s main body].

>In the revised version of our manuscript, we have carefully addressed your concerns as follows: (1.) We have introduced a dedicated analysis section and expanded the methodology and results sections, providing clearer details on the procedures, measures, and specific analyses that we’ve performed. (2.) We have expanded the theoretical framework section to provide a sharper foundation for our study. (3.) We have restructured the discussion section, now explicitly connecting our findings to prior research, outlining the study’s limitations, and suggesting directions for future work. We have integrated key information from the Supplementary Information into the main manuscript (as per your suggestion further below), ensuring that all essential details are directly accessible to the reader. Thank you for pointing this out.

_

2a. Introduction – General comments

To begin with, the introduction is too brief for a topic as crucial as the role of AI and the risks associated with trusting potentially risky financial investment advice. The research establishes part of the territory in the first paragraph (26 – 45) by explaining the importance of the study: three articles highlight the rising application of GenAI in financial investments by private investors (i.e., 1–3), while only one study provides concrete statistics related to the population in the UK (i.e., 4). Insufficient studies are presented, especially since they address the issues arising from the increasing reliance on GenAI. Moreover, while the literature may, let’s suppose, be scarce when it comes to the use of GenAI for financial advice, there might be relevant studies in other domains where overreliance can be detrimental and problematic. For instance, overreliance can have adverse effects in fields such as professional relationships, academic advice, etc.

>Agreed and done. We now provide a moderately expanded section on the overreliance on LLM advice and the adverse effects it can produce also in other fields. We believe that this somewhat broader introduction helped to set the stage why examining the impact on financial decision making is important. Thank you.

_

The second paragraph (47 – 58) sets out in more detail how GenAIs are prone to various biases, with several scientific articles supporting the point. However, the second paragraph ends by already presenting the research hypotheses and predictions. Indeed, the importance of research is established, and we have learned that more and more private investors are using GenAI advice and that GenAIs have significant biases. However, the paragraph does not establish the niche: we know neither whether there are counter-arguments to the use of GenAI for financial investments (i.e. that there may be benefits to its use) nor whether there is a gap in this issue, nor does it present questions that current research has not yet been able to answer. This paper does not show any limitations of previous research, how it can be developed further, or what it has missed. All these points are aimed at occupying the niche and giving a solution. Finally, the third paragraph (59 – 67) presents the results obtained right before detailing the method, somewhat echoing what was stated in the abstract. A brief overview of the ‘present study’ would have been appreciated instead.

>In the revised version of the paper, we now also highlight the benefits of LLM advice in the financial domain – at least for simple tasks such as doing a generic budgeting task. To the best of our knowledge, we are not aware of prior work that performed a causal test along the lines of the current work. This is now also better explained and highlighted in the revised overview section on “the present study”.

_

Finally, the introduction uses key terms yet fails to clarify or define them in detail. While finance experts may easily understand these terms as they have the necessary background, the scope of this research goes beyond this strict area. It should be accessible to individuals from other fields and, ideally, the general public. This research highlights the dangers of non-experts’ over-reliance on GenAI, non-experts who are not aware of those risks. Unfortunately, this article’s target audience may struggle to appreciate its implications fully. However, I must acknowledge that these terms are in Table 1, which contains nearly sufficient information to address and understand them. This table, only found in the middle of the Method section, summarises the measures, their definitions, their risks, and the calculations used for their accurate measurement. This table summarises definitions that should have been extensively defined beforehand, i.e., at the beginning of the theoretical background.

>Based on your comments and suggestions, we have now moved Table 1 (in the revised version of the paper Table 2) to the frontend of the paper and moderately expanded the explanation (and importance) of each of the main measures examined in the paper. We agree that featuring and explaining these measures earlier in the manuscript helps readers fully appreciate the focus and importance of this work. Thank you for helping sharpen to exposition already at the outset of the paper.

_

2b. Introduction – Small comments

l.48 Even if it is intuitive, is there any source that attests to this claim?

>We added two main citations to back-up this argument related to the inherent biases in large language models training corpora. Thank you for pointing this out.

_

l.43 – 45 These terms should have been defined beforehand to thoroughly understand their implications for the research. There is a risk of back-and-forth-reading between paragraphs.

>As highlighted earlier, and in response to your comments and suggestions, we have moved Table 1 (in the revised version of the paper Table 2) toward the frontend of the paper.

_

3a. Methods – General comments

In the beginning, I thought the mentioned factors of the risk tendency, the age, and the GenAI were conditions whose levels would be compared for each study in a comprehensive analysis, as it would have been done in a repeated-measures ANOVA or MANOVA (or linear mixed models with iterations as random effects). As mentioned in lines 72 to 76, we have the impression that a comparison between older and younger people would be made in the same manner as a comparison between the different levels of risk-taking (where higher risk-taking could supposedly lead to more significant biases), or even which GenAI would cause the riskier investments. No comparison between these levels is made in the Results section of the primary document. We then understand in the Results section that these variations in the prompts are used to give a more representative sample of prompts possibly used by private investors and that these are repeated over ten iterations each, i.e., 27 cross-conditions by 10 iterations for 270 occurrences in total. These iterations are presented as a factor in the experimental design: if it were the case, the levels of this factor would need to undergo pairwise comparisons (e.g., iteration 1 to iteration 2, iteration 1 to iteration 3, etc.). I would recommend presenting the interactions strictly as iterations of the three previous variables, not as a factor per se.

>Excellent point. We agree with your assessment and have revised the methodology section accordingly. Originally, we introduced iterations to account for potential selection bias when analyzing a single LLM response (Chen et al., 2023). In the revised version, we now explicitly present these repeated prompts as “iterations” and clarify our rationale for doing so. Additionally, we now first report a more traditional MANOVA model result with all investment risk measures as dependent variables and our experimental conditions (i.e., LLM type, risk-taking tendency, and age) as independent variables. To further explore individual-level effects, we then perform separate ANOVAs with post hoc Tukey HSD corrections for multiple comparisons for each investment risk, considering LLM type as the independent variable. We believe these revisions now follow a more standard approach in the context of reporting experimental data. Thank you for pointing this out.

_

Nevertheless, these comparisons between risk-tendency, age, and GenAI levels are carried out in the Supplementary Information (SI) but should be presented in the manuscript. Indeed, the SI has enormous potential for analyses that deserve to be presented in the manuscript itself.

>As noted in our previous response, we have now integrated the comparisons between risk-taking tendency, age, and LLM type in the results section of the manuscript. This ensures that key findings are immediately accessible to the reader.

_

Moreover, SI should be mentioned less in the manuscript. The manuscript should stand on its own as the article, so we do not have to read/refer many times to the SI.

>We fully agree that the manuscript should stand on its own, requiring fewer reference to the SI. To address this, we have incorporated additional details from the SI directly into the main manuscript, ensuring that key information is directly accessible to the reader. Wherever possible, we have expanded the manuscript to include relevant methodological and analytical details, reserving only supplementary analyses and extended discussions for the SI to maintain readability.

_

Finally, as we reach the conclusion of the section, there is no presentation of the analyses carried out (it seems there are several, in fact): which tests are used? how many? each measure one by one (several ANOVAs at the risk of increasing likelihood of alpha error)? a single and comprehensive analysis (e.g., MANOVA with 5 dependent variables)? which comparisons are carried out (Tukey HSD, LSD, Bonferroni,…)?

>To ensure greater clarity and transparency, we have now added a dedicated analyses section where we explicitly describe the statistical tests performed, including the number of analyses conducted and the rationale behind our approach. Additionally, we have expanded the results section to provide more detail on the specific analyses used, the structure of our comparisons, and the statistical corrections applied. We believe that these changes now better and more clearly communicate the analytical framework we used in the current research.

_

3b. Methods – Small comments

l.74-76 It is not clear who should invest in less risky assets, the younger or the older investors?

>Thank you for pointing this out. Originally, we stated that younger vs. older investors should generally invest in less risky assets due to older investors’ limited ability to recover from negative returns over their investment horizon (Barberis, 2000; Wachter, 2002). To clarify this point, we have expanded our explanation in the experimental paradigm section, ensuring a clearer discussion of how investment risk should be adjusted based on investor age.

_

l.104 TER refers to Total Expense Ratio, but the abbreviation was not introduced earlier. The financial term ‘Total Expense Ratio’ is not defined earlier (unless I am mistaken).

>Done. We have now introduced and defined the term “Total Expense Ratio (TER)” in the revised “Theoretical background & conceptual framework” section. Specifically, we clarify that “TER reflects the total costs associated with managing a fund, including management fees and administrative expenses, expressed as a percentage of assets invested.”

_

l.109 – 110 This builds on my earlier point regarding content that might not be understandable to those unfamiliar with the field. The abbreviation ‘FRED’ is neither introduced nor explained, leaving me – and/or others – unable to grasp the sentence fully.

>Done. This is now explained in the same section (i.e., FRED is the online database of the Federal Reserve Bank that lists the “risk-free market rate”).

_

4a. Results – General comments

In general, the studies’ results are well presented. We know where the comparisons are significant and where GenAI is a problem when compared to the benchmark. The results are also interesting: even with the investment disclaimer avoiding bias (Study 2a, 2b, 3), some of them still stand out. However, I recommend compiling all these descriptive statistics into a summary table for easier readability and to allow you more room to elaborate on each result, such as which sectors or geographical areas are over- or under-represented.

>We appreciate your thoughtful feedback regarding the presentation of our results. We agree that the extensive statistical details in the main text reduced readability and limited space for more nuanced exploration of findings. Following your suggestion, we have compiled comprehensive summary tables for all ANOVAs conducted across our studies. To maintain accessibility for PLOS ONE’s broad readership, we have placed these detailed statistical tables in the appendix while retaining the more intuitive figures in the main text. This approach preserves the visual representation of key findings in our initial submission, while ensuring that all statistical analyses are thoroughly documented and available for interested readers. We are open to include these tables in the main manuscript if you believe this is absolutely necessary. We decided to move the detailed contrasts to the appendix to better address your comments and suggestions to improve the readability of the results section and we hope we did so to your satisfaction

_

I found the

---

## [Decision Letter · Decision Letter 1]

Biased echoes: Large language models reinforce investment biases and increase portfolio risks of private investors

PONE-D-24-45837R1

Dear Dr. Winder,

We’re pleased to inform you that your manuscript has been judged scientifically suitable for publication and will be formally accepted for publication once it meets all outstanding technical requirements.

Kind regards,

Peter Gordon Roetzel

Academic Editor

PLOS ONE

Additional Editor Comments (optional):

Reviewers' comments:

Reviewer's Responses to Questions

**Comments to the Author**

Reviewer #2: All comments have been addressed

Reviewer #3: All comments have been addressed

2. Is the manuscript technically sound, and do the data support the conclusions?

Reviewer #2: Yes

Reviewer #3: Yes

3. Has the statistical analysis been performed appropriately and rigorously?

Reviewer #2: Yes

Reviewer #3: Yes

4. Have the authors made all data underlying the findings in their manuscript fully available?

Reviewer #2: Yes

Reviewer #3: Yes

5. Is the manuscript presented in an intelligible fashion and written in standard English?

Reviewer #2: Yes

Reviewer #3: Yes

Reviewer #2: i would like to thank the authors for the efforts been made on this article. I can see that the comments been addressed, so the paper now is ready to be published.

Reviewer #3: The investigation of "Biased echoes: Large language models reinforce investment biases and increase portfolio risks of private investors" emphasises, again, that technical tools must be used by professionals; a chainsaw should never be used by an amateur; a very interesting study indeed.

In responses to the earlier review, I can see that a lot of effort has been put in to address the identified issues. While any study cannot be 'perfect', the question is whether the study adds to our understanding of something; in reading this study, I am convinced that it definitely does. Since LLMs are built on human generated studies, these will surely will have the same biases - refuting the overzealous claims from the developers of statistical inference tools termed as AI.

**Do you want your identity to be public for this peer review?** For information about this choice, including consent withdrawal, please see our Privacy Policy

Reviewer #2: No

Reviewer #3: No

---

## [Editor Report · Acceptance letter]

PONE-D-24-45837R1

PLOS ONE

Dear Dr. Winder,

I'm pleased to inform you that your manuscript has been deemed suitable for publication in PLOS ONE. Congratulations! Your manuscript is now being handed over to our production team.

Kind regards,

on behalf of

Dr. Peter Gordon Roetzel

Academic Editor

PLOS ONE